# Optimising CNT-FET biosensor design through modelling of biomolecular electrostatic gating and its application to β-lactamase detection

Rebecca E. A. Gwyther[1,6], Sébastien Côté [2,3,6] ✉, Chang-Seuk Lee [4,5,6], Haosen Miao[4], Krithika Ramakrishnan[1], Matteo Palma [4] ✉ & D. Dafydd Jones [1] ✉

Carbon nanotube field effect transistors (CNT-FET) hold great promise as next generation miniaturised biosensors. One bottleneck is modelling how proteins, with their distinctive electrostatic surfaces, interact with the CNT-FET to modulate conductance. Using advanced sampling molecular dynamics combined with non-canonical amino acid chemistry, we model protein electrostatic potential imparted on single walled CNTs (SWCNTs). We focus on using β-lactamase binding protein (BLIP2) as the receptor as it binds the antibiotic degrading enzymes, β-lactamases (BLs). BLIP2 is attached via the single selected residue to SWCNTs using genetically encoded phenyl azide photochemistry. Our devices detect two different BLs, TEM-1 and KPC-2, with each BL generating distinct conductance profiles due to their differing surface electrostatic profiles. Changes in conductance match the model electrostatic profile sampled by the SWCNTs on BL binding. Thus, our modelling approach combined with residue-specific receptor attachment could provide a general approach for systematic CNT-FET biosensor construction.

Nanocarbon devices such as carbon nanotube field effect transistors (CNT-FETs) are rapidly becoming the base biosensing material of choice due to their desirable electronic properties and routes to functionalisation[1–4] enabling label-free detection of biomolecules with high sensitivity to specific targets using miniaturised devices[5–10]. In a CNT-FET biosensor setup, biomolecular interaction events are detected due to the electrostatic field generated by the biomolecules in proximity of the CNT surface inducing changes to charge carrier density and thus modulating CNT conductance[2,6,11]. To enable target specificity, single walled CNTs (SWCNTs) are typically decorated with receptors (e.g. DNA aptamers[12–16] or binding proteins[17–21]) that

recognise and bind the required analyte. In this regard, protein-protein interactions (PPIs) are of particular interest as they are widespread in nature, being central to a range of biological events and are at the heart of modern diagnostics.

Proteins have surfaces comprised of basic (positive) and acidic (negative) residues giving rise to an electrostatic surface unique to each protein. Whilst there have been notable successes of proteins electrostatically gating SWCNTs[17–19], their effectiveness is generally limited by how the receptor is attached to the SWCNT surface. Ideally, we should attach the receptor in a defined manner (i.e. at a predefined site), with an optimal orientation to impart the greatest electrostatic

[1]Molecular Biosciences Division, School of Biosciences Cardiff University, Cardiff, UK. [2]Département de Physique, Faculté des Arts et des Sciences, Université de Montréal, Montréal, QC, Canada. [3]Département de Physique, Cégep de Saint-Jérôme, Saint-Jérôme, QC, Canada. [4]Department of Chemistry, Queen Mary University of London, London, UK. [5]Department of Chemistry, Seoul Women's University, Seoul, Republic of Korea. [6]These authors contributed equally: Rebecca E. A. Gwyther, Sébastien Côté, Chang-Seuk Lee. ✉e-mail: scote@cstj.qc.ca; m.palma@qmul.ac.uk; jonesdd@cardiff.ac.uk

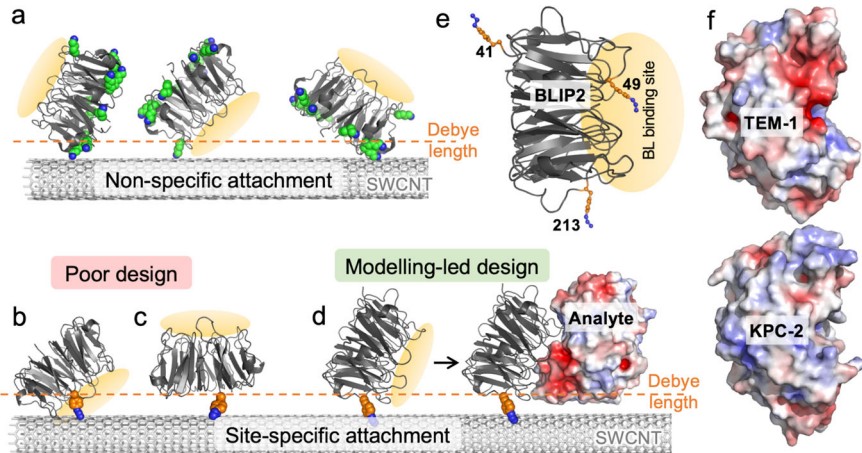

**Fig. 1 | Outline of the CNT-FET design. a** Examples of non-specific receptor protein (BLIP2) attachment via primary amines with lysine residues shown as green spheres. The Debye length is shown as a dashed orange line. **b, c** Examples of poor design for site-specific attachment whereby the BLIP2 binding interface is blocked by the SWCNT (**b**) or binding of the analyte will be well beyond the Debye length (**c**). **d** Example of an ideal, modelling-led design whereby the analyte binding is within the Debye distance and imparts an electrostatic profile on the SWCNT. **e** Structure of BLIP2 highlighting the BL binding interface (orange oval) and selected interface residues (orange sticks) mutated to azF. **f** Different surface electrostatic profile of two BLs, TEM−1 (top) and KPC-2 (bottom). The equivalent surfaces are shown for each. Red and blue represent acidic and basic patches, respectively.

effect onto the SWCNT upon analyte binding. However, most receptor protein attachment is not controlled and is essentially random leading to heterogenous orientations of the receptor (and in turn the analyte), of which many are far from optimal (see Fig. 1a for examples), generating signal variation from device to device, and even signal cancelling effects. There are some notable successes regarding designed attachment via specified residues, usually cysteine thiol chemistry[18,19] or more recently non-canonical amino acid chemistry incorporated using a reprogrammed genetic code approach[17,22,23]. The latter is particularly powerful choice as there is a broader selection of coupling chemistry available and can also allow attachment either directly to the SWCNT[22,23] or via an intermediate linker[17,24,25]. For example, genetically encoded phenyl azide chemistry[26] enables direct photochemical covalent attachment[23] or linker-based click chemistry[17,25] approaches to be used.

The main issue with residue-specific attachment is picking the right residue to maximise biosensing potential so as to avoid: (1) steric clashes between receptor protein and/or analyte protein and the SWCNT; (2) no analyte binding due to masking of the receptor binding site (Fig. 1b); (3) analyte binding far beyond the Debye distance (Fig. 1c). The Debye length is particularly important as it defines the distance from the SWCNT where the electrostatic influence is greatest[16,17,27]. Clearly, modelling how a receptor protein is attached and how the subsequent analyte binds with respect to the SWCNT have the potential to unveil vital information at the atomistic scale regarding biosensing potential, thereby guiding the development of optimal CNT-FET designs (Fig. 1d). Despite this, to the best of our knowledge, modelling has not yet been leveraged together with site-specific protein attachment strategies to develop CNT-FET biosensors.

Herein we focus our efforts on the detection of a group of enzymes, β-lactamases (BLs), that are one of the main causes of antimicrobial resistance (AMR)[28–30] Class A serine β-lactamases secreted by bacteria, such as the clinically important TEM-1[31] and KPC-2[32], are the main causes of resistance to β-lactam antibiotics so their quick detection during infection could prove vital in administering the right antibiotic therapy. The β-lactamase inhibitory protein 2 (BLIP2; Fig. 1e)[33,34] is a near "universal" binder for class A BLs so acts as the ideal receptor protein for detecting a broad range of BLs, including TEM-1[35] and KPC-2[36]. BLIP2 recognises the 3D arrangement of the BL active site using a circular arrangement of loop structures (Fig. 1e)[35]. While TEM-1 and KPC-2 are structurally (Cα root-mean-square deviation of 0.81 Å)

and functionally similar, they have distinctive electrostatic surface profiles (Fig. 1f) due to the differences in the number and distribution of charged amino acids comprising each BL.

By combining computational modelling approaches, genetic code reprogramming, chemical biology, and nanoscale fabrication of biosensing devices, we successfully photochemically attach BLIP2 at defined, designed residues, enabling the detection of TEM-1 and KPC-2; each BL generates distinct conductance profiles due to differences in their unique surface electrostatic patches presented close to the SWCNT surface; the changes in conductance closely matching the model electrostatic profiles sampled by the SWCNTs on BL binding.

## Results and discussion

### Modelling receptor protein attachment site feasibility

We have recently developed an approach whereby a protein can be precisely and intimately interfaced directly with nano-carbon, including SWCNTs, using genetically encoded phenyl azide chemistry via the non-canonical amino acid azF (4-azido-L-phenylalanine)[22,23,37]. Compared to pyrene-based attachment methods that require an extended linker (Supplementary Fig. 1)[17], direct phenyl azide photochemical insertion provides a more intimate connection between receptor protein (and thus analyte protein) and the SWCNT surface potentially allowing for more efficient signal transduction. Not only will pyrene-based linking approaches place the receptor further from the SWCNT but rotation around bonds within the linker increases flexibility resulting in higher receptor protein conformational heterogeneity with respect to the SWCNT; direct phenyl azide photochemical attachment should generate a defined receptor-SWCNT interface orientation allowing more precise modelling and device construction. Here we aimed to devise an accurate method to model the likely initial binding orientation of the receptor protein allowing us to estimate the distance and thus electrostatic influence between the incoming analyte protein and the SWCNT. BLIP2 will act as the receptor protein on the CNTs, for both its potential in antimicrobial resistance (AMR) diagnostics, and to test our modelling approach. As BLIP2 binds a range of different BLs, this allowed us to model how different analyte proteins with distinctive electrostatic surfaces influence SWCNTs and thus conductance.

The first step is to assess whether a particular attachment residue on the receptor protein [BLIP2] is viable for interfacing with a SWCNT by using molecular dynamics to sample rotamer forms of azF. We then used a more extensive electrostatic profiling sampled by the SWCNT

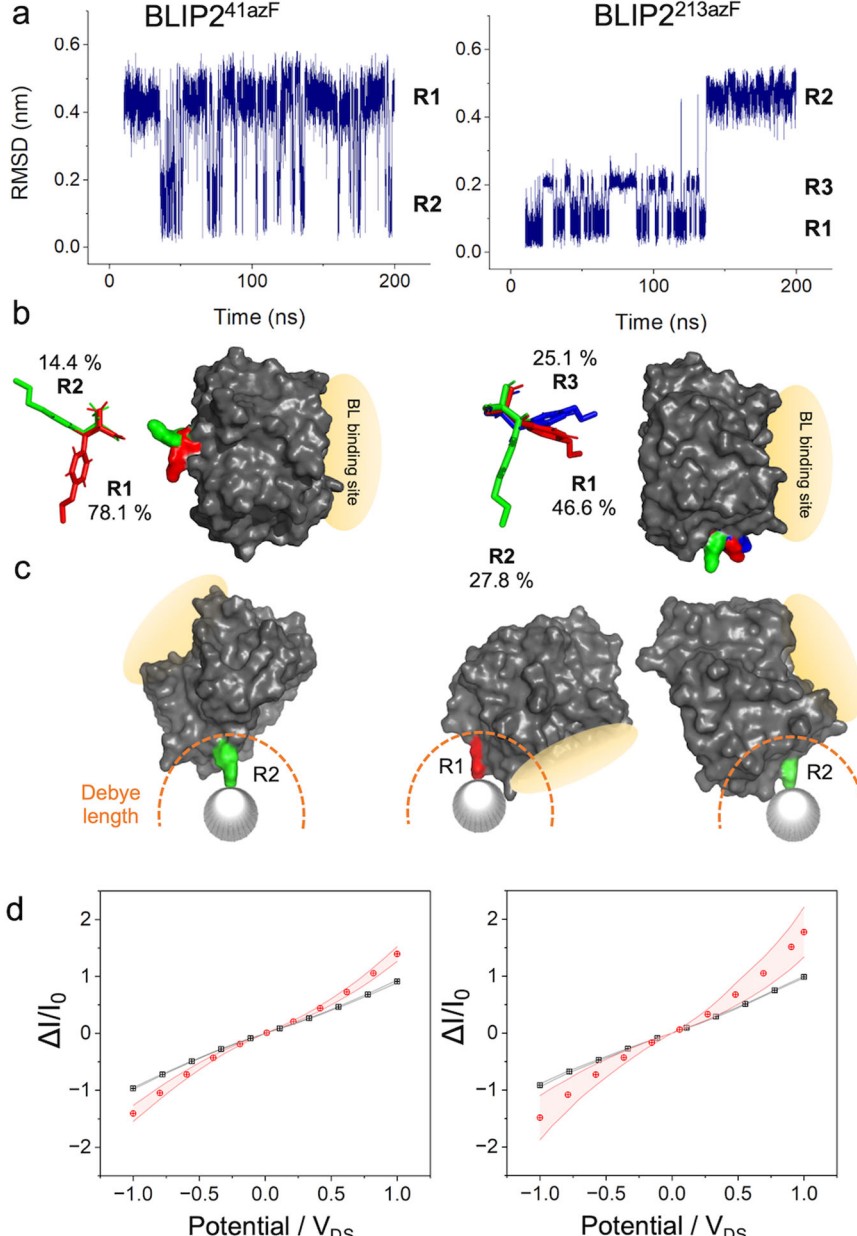

**Fig. 2 | Modelling of BLIP2 mutants (azF41 left column and azF213 right column) azF sidechain rotamer configurations and SWCNT binding. a** Root mean squared deviation (RMSD) for the azF residue over one simulation repeat (190 ns) in BLIP2$^{41azF}$ and BLIP2$^{213azF}$, with rotamer states identified from step changes in RMSD. **b** Modal rotamer configurations were extracted and compared across all repeats ($n = 3$) to give an average rotamer propensity, labelled as percentage of total simulation time. **c** Viable rotamers for SWCNT docking. Docking of BLIP2 azF mutant onto the SWCNT is based on the geometry of the known [2 + 1]

cycloaddition reaction, whereby the nitrene radical formed on exposure to UV light inserts perpendicular to the SWCNT[41]. Rotamers were deemed viable if no steric clash occurred between BLIP2 and the SWCNT. The BL binding site is highlighted in yellow and Debye length is depicted in orange. **d** I/Vs before (black) and after (red) BLIP2 azF attachment; BLIP2$^{41AzF}$ ($n = 12$) and BLIP2$^{213azF}$ ($n = 8$). Standard error of the mean is plotted either side of the average measurement values. Source data for plots in (**a**) and (**c**) are provided in the Source Data file.

on analyte protein [the BLs] binding to assess the electrical response of the CNT-FET. Structures available in the Protein Data Bank (PDB) (e.g. BLIP2-TEM-1 complex[35]) or models generated by in silico methods such as AlphaFold2 (AF2) multimer[38–40] (e.g. BLIP2-KPC-2; Supplementary Fig. 2) can be used as the starting point. The selected BLIP2 residues are then mutated in silico to incorporate azF and the BLIP2 receptor variant structures are used as starting points for molecular dynamics to sample the rotamer configurations of the azF side-chain. We selected two BLIP2 residues, 41 and 213 (Fig. 1b), as models to test our approach. We have shown previously that site-specific attachment via

these residues through an intermediary pyrene linker allowed detection of TEM-1 in a CNT-FET setup[17].

MD simulations suggested that BLIP2 with azF at residue 41 (BLIP2$^{41azF}$) sampled two major rotamer conformations covering 92.5% of the sampled frames (Fig. 2a, b). Docking of rotamer 1 (R1) onto a SWCNT suggested significant steric clashes between the protein and nanotube (Supplementary Fig. 3). No steric clashes were observed when rotamer 2 (R2) was manually docked to the SWCNT. While R1 was the less populated of the two major rotamer forms, it is sampled on a regular basis during the simulation (Fig. 2a). Docking of TEM-1 onto R2

BLIP2[41azF]-CNT suggested that the analyte BL would still bind to the receptor-CNT complex (Fig. 2c).

BLIP2 with azF at residue 213 (BLIP2[213azF]) accessed 3 main rotamer conformations (Fig. 2a, b) accounting for 99.5% of the frames. Only rotamer 3 (R3) displayed steric clashes when manually docked on the SWCNT (Supplementary Fig. 3). On docking BLIP2[213azF] R1 and R2 forms to the SWCNT, both rotamer forms retained the ability to bind TEM-1. This initial modelling suggested that R1 brought the BL close to the SWCNT surface while R2 caused the BL binding site of BLIP2[213azF] to project away from the nanotube (Fig. 2c).

We then experimentally confirmed that both BLIP2[41azF] and BLIP2[213azF] could be photochemically attached to SWCNT bundles in the CNT-FET setup. We have previously shown that BLIP2[41azF] could be attached to single SWCNTs by AFM[23], which we confirmed here with SWCNT bundles for both BLIP2[41azF] and BLIP2[213azF] (Supplementary Fig. 4 and associated discussion). Additional height increases were observed on the addition of TEM-1 to the same SWCNT bundles (Supplementary Fig. 4a–c) and when monitoring single SWCNT protrusions (Supplementary Fig. 4d, e) providing evidence that BL binding capacity is retained (and thus BLIP2 remains folded after attachment) as the initial modelling suggested. From AFM data analysis where single SWCNTs could be observed (see Supplementary Fig. 4d for an example), we estimate that AzF containing BLIP2 bound with a density of 3.6 ( ± 0.7) proteins per 100 nm per SWCNT. Photochemical BLIP2 azF attachment was also confirmed by I/V measurements in a CNT-FET setup using p-type semiconducting SWCNTs (Fig. 2d). The I-V measurements pre-functionalisation display ohmic behaviour; with a linear relationship between current and voltage, as expected with pristine SWCNTs. Covalent attachment of either BLIP2 azF variants resulted in increased conductance in line with the modelling that suggested both attachment sites impose a negative electrostatic potential on the SWCNT (vide infra). The introduction of breaks into the SWCNT sp² bond network would normally lead to a reduction in conductance; by using the phenyl azide photochemical approach, our work suggests that this can be offset, with recent observations indicating aromatic azides maintain the π electron network upon covalent functionalisation[41].

### Modelling electrostatic effect of protein on SWCNT

The ability to model the electrostatic influence proteins have on the SWCNT conductance channel is critical to the successful design of FET-based biosensors. Here we have developed an approach to model electrostatic effects of protein binding to a SWCNT. The approach used: (1) exhaustively sampling the orientations of receptor protein with respect to the carbon nanotube using all-atom, solvent explicit Hamiltonian-replica exchange molecular dynamics (H-REMD) simulations[42]; and (2) determining the electrostatic potential generated by these orientations on the surface of the SWCNT using electrostatic Poisson-Boltzmann simulations[43]. Using this procedure, we quantified the electrostatic change due to attachment of BLIP2 via azF41 or azF213 to the SWCNT and the subsequent association of TEM-1 and KPC-2. From the electrostatic potential, we then infer the impact on the charge carrier density in the SWCNT and thus on its conductance.

First, the 1250 ns per replica H-REMD simulations thoroughly sampled a variety of orientations that progressively converged towards the most thermodynamically stable orientations at the unscaled energy (Supplementary Fig. 5). Over the last 750 ns, BLIP2[41azF] and BLIP2[213azF] sample four main orientation ensembles (Supplementary Fig. 6). Inspection of the centroid of these ensembles reveals that TEM-1 is localized very differently with respect to the nanotube when associating to BLIP2[41azF] compared to BLIP2[213azF]. Indeed, while all residues of TEM-1 are farther than 1.0 nm from the nanotube for BLIP2[41azF], many charged residues are within 1.0 nm of the nanotube for BLIP2[213azF] (Supplementary Fig. 7). Consequently, we expect that the electrostatic

potential change on the nanotube will be significantly stronger when the analyte (TEM-1) binds to the BLIP2[213azF] as the receptor.

Next, we quantified the electrostatic potential (ESP) change by solving the Poisson-Boltzmann equation for the configurations sampled from the H-REMD simulation for each attachment site. Binding of either BLIP2[41azF] or BLIP2[213azF] alone imparts a largely negative ESP on the SWCNT (Supplementary Fig. 8), in line with increased conductance of p-type SWCNTs observed in Fig. 2d. Binding of TEM-1 elicits an average reduction of the ESP on the order of 1 mV on the nanotube for BLIP2[41azF], while the reduction is an order of magnitude larger for BLIP2[213azF] (10 mV) (Fig. 3a). To quantitatively compare the two sites, we calculated the global ESP change on the surface of the nanotube (Fig. 3b). Binding of TEM-1 produces a relatively small negative global ESP change with a main peak at −2463 ± 719 a.u for BLIP2[41azF]; this is significantly stronger for BLIP2[213azF] with two main peaks at −9526 ± 2543 a.u and −26,871 ± 6207 a.u.

As shown in Fig. 1f, KPC-1 has a distinct electrostatic surface profile to TEM-1, with KPC-2 having an overall charge of −1 (26 negatively and 25 positively charged residues), while TEM-1 is −7 (36 negatively and 27 positively charged residues). Our simulations indicate that this difference in surface charge profile for KPC-2 has a strong impact on the ESP imparted on the nanotube, particularly for BLIP[213azF] for which the ESP profile becomes positive instead of being negative (Fig. 3a). In term of overall ESP, two peaks are again observed for KPC-2 on binding BLIP2[213azF] but now with a significant shift to a positive ESP sampled by the SWCNT (25,915 +/− 3708 and 8482 +/− 3424 a.u.; Fig. 3b); association of KPC-2 results in a relatively small negative overall ESP on binding BLIP2[41azF] (−799 +/− 369 a.u.).

Thus, our modelling suggests that TEM-1 binding to BLIP2[213azF] is likely to exert the greatest ESP on SWCNT; the increased negative charge close to the p-type SWCNT conductance channel should increase the positive hole carrier density and so increase conductance[16,17,44,45]. TEM-1 binding to BLIP2[41azF] should only exert a small negative ESP so should generate a relatively small increase in conductance. In comparison, KPC-2 binding to BLIP2[213azF] will impart a significant positive charge on the p-type SWCNT so reducing conductance. Binding of KPC-2 to BLIP2[41azF] should exert a small negative ESP which will potentially have a very minor effect on conductance.

### Conductance of designed BLIP2 receptor CNT-FETs

We next tested our models to see if conductance changed in the anticipated manner in a CNT-FET setup. All CNT-FETs used were comprised of p-type SWCNTs, and I-V traces were measured on addition of the receptor [BLIP2; Fig. 2d] and then analyte [BL] (Fig. 4a).

Changes in conductance were observed on addition of BL to BLIP2 modified devices indicating productive analyte-receptor interactions, which is only possible if BLIP2 retains its native structure on SWCNT attachment. Notably, on addition of a BL, the I-V traces for BLIP2[213azF] devices gave the largest changes in conductance compared to devices with BLIP[41azF] (Fig. 4b), in line with the ESP models. Additionally, the BLIP2[213azF] devices behaved differently depending on the BL added. Addition of TEM-1 resulted in increased conductance whereas KPC-2 caused conductance to decrease (Fig. 4b). This matches our modelling whereby TEM-1 binding to BLIP2[213azF] exerts a negative ESP on the SWCNT (Fig. 3b) so enhancing positive hole charge carrier mobility our p-type devices resulting in increased conductance. KPC-2 on the other hand exerts a positive ESP on the SWCNTs which will decrease conductance.

TEM-1 binding to BLIP2[41azF] resulted in only a small conductance increase (Fig. 4b and Supplementary Fig. 9), 4-fold lower than that observed for BLIP2[213azF] devices. Modelling suggests that TEM-1 binding to BLIP2[41azF] exerts a relatively weak negative ESP profile on the SWCNT, so our experimental results are consistent with the models (Fig. 3b). KPC-2 exerts an even weaker ESP on SWCNTs, and this would

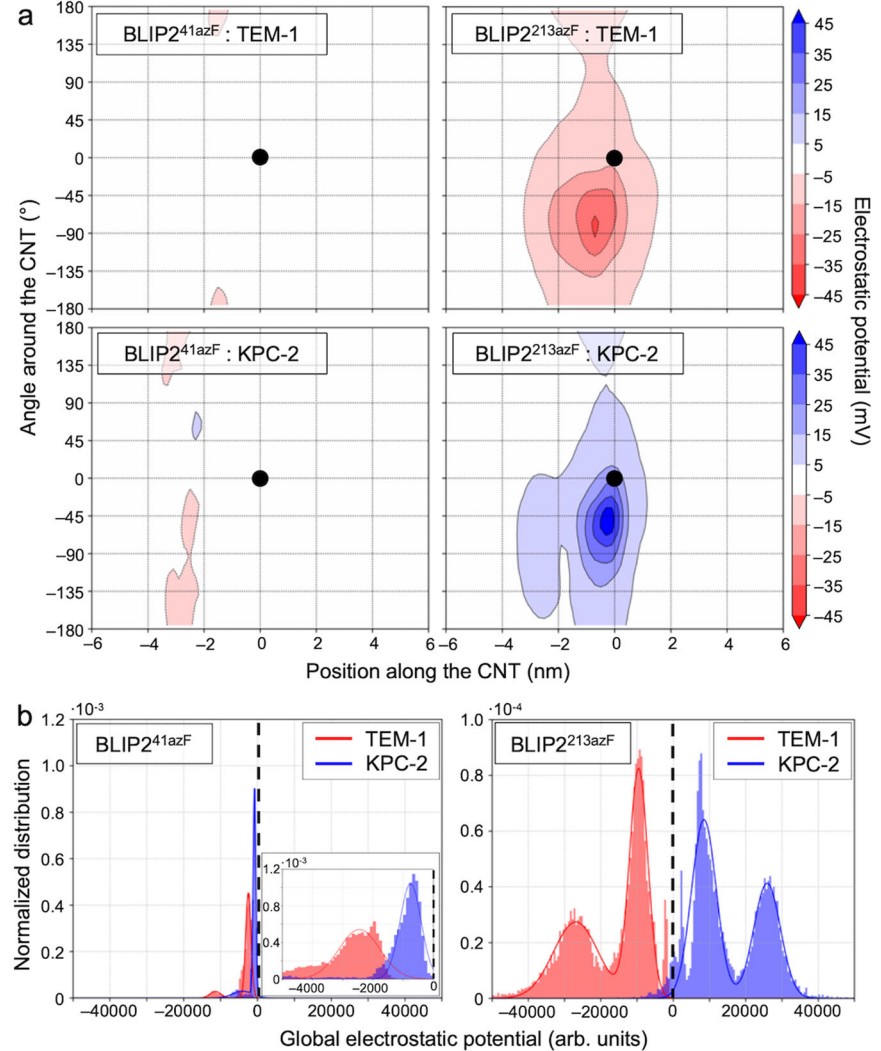

**Fig. 3 | Change of electrostatic potential on analyte binding.** The change of electrostatic potential (ESP) on the carbon nanotube due to the association of TEM-1 and KPC-2 with BLIP2$^{41azF}$ (left) and BLIP2$^{213azF}$ (right). The BLIP2 alone ESP profiles are shown in Supplementary Fig. 8. **a** The average electrostatic potential exerted by each BL as a function of the position along the nanotube axis and of the angle around the nanotube where the coordinate (0,0) corresponds to the attachment site (black dot). **b** The normalized histogram of the global ESP of each BL on the nanotube. The global electrostatic potential is computed using a Riemann sum for each sampled configuration separately. The black dashed line indicates no ESP change compared to BLIP2 alone. For BLIP2$^{41azF}$ binding a BL, inset is the magnified region of the global electrostatic potential. **a**, **b** The statistics are performed on the electrostatic potential maps of the sampled configurations in the 500 to 1250 ns interval of the H-REMD simulations (see Methods).

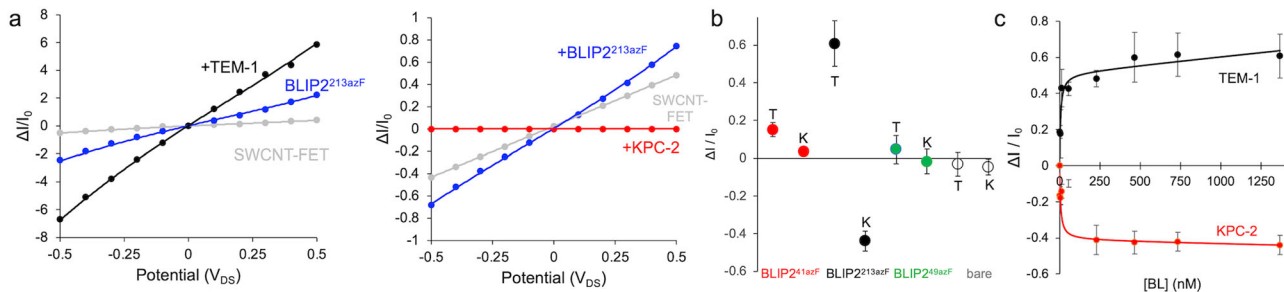

**Fig. 4 | Conductance measurement with BLIP2 azF variants photochemically attached to CNT-FET devices upon the addition of β-lactamase. a** Example I-V traces for BLIP2$^{213azF}$ binding TEM-1 (left) and KPC-2 (right). The plots represent bare SWCNT-FET (grey), attachment of BLIP2$^{213azF}$ (blue), addition of 1300 nM TEM-1 (blue, left), and addition of 1300 nM KPC-2 (red, right), respectively. **b** Relative change in conductance on addition of either TEM-1 (T) or KPC-2 (K) to either bare SWCNT-FETs (clear) or with BLIP2$^{41azF}$ (red), BLIP2$^{213azF}$ (black), BLIP2$^{49azF}$ (green) acting as the SWCNT bound receptor. Each BL was added to a final concentration of 1.4 μM. Data is presented as the mean value +/− the SD ($n = 3$, where n is the number different devices measurements were derived from). **c** BL concentration-dependent change in current (at $V_{DS}$ of 0.1 V) plots for BLIP2$^{213azF}$. TEM-1 and KPC-2 conductance profiles are coloured black and red, respectively. The conductance data were collected via I-V measurements between source-drain. The data points where fit to one site binding equation in GraphPad Prism. Data is presented as the mean value +/- the SD ($n = 3$, where n is the number different devices measurements were derived from). Source data for plots in (**a**–**c**) are provided in the Source Data file.

explain the negligible change conductance (Fig. 4b and Supplementary Fig. 9).

The conductance profiles observed here differ from those observed previously using a pyrene-linker-dibenzylcyclooctyne (PLD) system to interface BLIP2 via azF41 or Az213 using click chemistry[17]. This is largely expected as both the length and conformation flexibility of the PLD will result in different distances and orientations of BLIP2 with respect to the SWCNT sidewall (and thus of the incoming BL analyte) compared to the rigid, zero length linker approach used here (see Supplementary Fig. 1 for comparison). Thus, the nature of the attachment process as well as the specific residue is important for defining device characteristics.

We also used a BLIP2 variant, BLIP2[49azF], as a control receptor that is known to sterically block BL binding on SWCNT attachment[17]; the azF photochemical attachment handle is within the BL binding region (Fig. 1e). As shown in Fig. 4b (and Supplementary Fig. 10a), little change in conductance is observed on addition of either BL to BLIP2[49azF] CNT-FETs, as expected. Furthermore, each BL does not induce significant changes in conductance of bare SWCNTs highlighting the that little non-specific protein binding to the CNT surface itself is occurring and the importance of the receptor (Fig. 4b and Supplementary Fig. 10b).

We next investigated BL concentration dependent changes in conductance for each device setup. Initially Dulbecco's phosphate buffered saline (DPBS) buffer was added to the channel region followed by stepwise additions of the BLs, with I-V curves measured after every BL addition (Fig. 4c, Supplementary Fig. 9 and Supplementary Fig. 10). For the BLIP2[213azF] receptor devices, we observed a TEM-1 concentration dependent change in conductance (Fig. 4c). Even at low TEM-1 concentrations (sub 100 nM) close to maximal signal was observed suggesting saturation of BLIP2 binding sites and retention of high affinity binding observed for native BLIP2 to BLs[46,47]. As noted above, we did see a smaller increase in conductance for TEM-1 binding BLIP2[41azF] which did show a concentration dependence (Supplementary Fig. 9). For KPC-2, a similar concentration dependence was observed with BLIP2[213azF] CNT-FETs, but with a decrease in conductance now used to monitor BL binding (Fig. 4c). Again, signal saturation was reached at relatively low concentrations of KPC-2 suggesting that BLIP2[213azF]'s high BL affinity was retained. No clear KPC-2 binding signal was observed with BLIP2[41azF] modified CNT-FETs (Supplementary Fig. 9) suggesting there was no ESP change on the SWCNT. Moreover, both bare SWCNTs and BLIP2[49azF] showed no BL concentration-dependent signal change (Supplementary Figs. 10a, b, respectively), nor did the addition of bovine serum albumin (BSA) affect the conductance behaviour of devices functionalised with BLIP2[213azF] (Supplementary Fig. 10c), ruling out non-specific binding effects. Notably, in the presence of excess BSA as an interference agent, BLIP2[213azF] had a similar conductance profile on addition of TEM-1 as in the absence of BSA (Supplementary Fig. 10d). While we cannot entirely exclude that Schottky-barrier modification[2], potentially induced by non-specific protein adsorption, may play a role in our system, it is reasonable to assume that an electrostatic gating effect induced by the specific recognition of BLs by BLIP tethered to SWCNTs, is significantly contributing to the sensing response of our detection platform.

In conclusion, while the use of a bioCNT-FET setup offers great potential for biosensing and even in molecular electronics, it is of paramount importance to optimise the device design process. The standard route of non-specific attachment of protein receptors leads to a highly heterogenous system in terms of receptor orientation, and thus analyte binding that may even result in electrostatic potential effects cancelling each other out, so impairing the optimal performance of the biosensor. Here we show in silico modelling has the potential to predict accessible receptor attachment sites and the likely ESP sampled by a SWCNT, which acts as the transducer in FET biosensors that manifests as expected conductance changes. Critical is

the use of a non-natural amino acid that restricts the coupling site to a single defined residue on the receptor providing a homogenous and precise link to the SWCNT, and so defining the modelling process. Moreover, we show that similar proteins but with differing surface ESP profiles elicit discrete conductance profiles, an aspect that we can also predict through modelling. Here we show that two different BLs with differing clinical effects in terms of their resistance profile generated different conductance profiles. Combining multiplexing with a suite of BLIP2s engineered to present different BL electrostatic surfaces could pave the way for rapid sensing of BLs based on their electrostatic profiles present during infection leading to more appropriate antibiotic prescription regimes. We envisage that our approach can be used more broadly allowing the generation of optimised protein receptor attachment for improved analyte sensing. Existing structures or accurate models of receptor-analyte complexes generated in silico by, for example AF2 multimer[38–40], can act as the input for initial systematic modelling to allow quick identification of permissible attachment sites. These selected sites then act as the input for more detailed molecular dynamics to identify ideal attachments sites for experimental analysis, with azF introduced into the receptor at the desired residue through established protein engineering and recombinant protein production approaches. Ultimately, a streamlined process could be developed whereby simply supplying the receptor-analyte sequence will generate design sets for optimal bioFET construction for researchers to implement. Our modelling approach will also provide an invaluable aid to understanding the likely spatial and functional relationship of protein-bionanohybrid systems that helps us to understand fundamental basis of action.

## Methods
### Modelling BLIP2 azF mutations and rotamers
Input structures were prepared using AlphaFold v2.1.0[39] using the translated gene sequence of BLIP2, to most accurately represent the protein recombinantly produced. The structures produced are highly similar to the known crystal structures of BLIP2 with an all atom RMSD of 0.461 and 0.455 Å for BLIP2[41azF] and BLIP2[213azF], respectively, compared to PDB entry 3qi0[47]. The azF mutation was introduced manually using the SwissSidechain plugin in PyMOL[48]. All simulations were performed in GROMACS[49,50] with the CHARMM36 forcefield[51]. The forcefield was modified to add in the new AzF residue, with partial charges derived from bond charge corrections using ACPYPE[52]. BLIP2 was placed in a dodecahedron box, solvated with TIP3P water and neutralising sodium ions. Energy minimisation was performed with the steepest descent algorithm of maximum 50000 steps and convergence below 1000 kJ mol$^{-1}$ nm$^{-1}$. Long-range electrostatics were determined by Particle-mesh Ewald (cutoff of 1.2 nm) and periodic boundary conditions (PBC) were applied. Temperature and pressure equilibration used the leap-frog algorithm to equilibrate the solvent about the protein. Velocity-rescale temperature coupling and isotropic Berendsen coupling maintained temperature and pressure at 300 K and 1 bar using a coupling constant of 100 ps. These parameters were taken forward for the production run using a 2 fs timestep, totalling 200 ns, with energies and coordinates being collected every 10 ps. Simulations were performed in triplicate for each protein. Post-MD, BLIP2 was centred in the box as treatment for PBC and fitted to the reference file by translation. Structural stability analysis was performed to test the viability of the model, and the first 10 ns were trimmed from all trajectories to discount the equilibration phase of the simulation.

### H-REMD simulations
HREX simulations were performed using the open-source software GROMACS (2019.6) augmented with the open source patch PLUMED 2.6.2. Electrostatic computations were performed using the open-source software APBS 1.3. Parametrization of the linker was done using

Antechamber 21.0 as well as ACPYPE. All-atom, solvent explicit Hamiltonian replica-exchange molecular dynamics (H-REMD) simulations were performed on BLIP2 covalently attached to a carbon nanotube for the two attachment sites (BLIP2[41azF] and BLIP2[213azF]). Below, we briefly present the simulated systems with more detailed descriptions in the SI.

The protein is modelled using the AMBER14sb force field[53]. The attachment between BLIP2 and the nanotube is parametrized using the Generalized AMBER forcefield (GAFF) and Restrained electrostatic potential (RESP) procedures from AmberTools20 to respectively determine the AMBER atom types (bonded and Lennard-Jones parameters) as well as the partial charges of the attachment (Supplementary Fig. 11 and SI for more details)[54]. We used a (10,0)-nanotube (one of the major chiralities present in our CNT mixture) with atoms modelled as uncharged sp2 carbon of type CA in AMBER, with Lennard-Jones parameters $\sigma = 0.339967$ nm and $\varepsilon = 0.359824$ kJ/mol, and their position is frozen during the simulation. Water molecules and ions are respectively modelled using TIP3P and Joung-Cheatham parameters[55].

The H-REMD simulations were performed using the GROMACS software[50,56] version 2019.6 augmented with the open-source, community-developed PLUMED library[57,58] version 2.6.2 for H-REMD simulations[59]. The simulations were done in the NVT ensemble using the Bussi-Donadio-Parrinello thermostat[60] with a reference temperature of 300 K. Bonds involving a hydrogen atom were constrained using the P-LINCS algorithm and the geometry of the water molecules was constrained using the SETTLE algorithm, allowing an integration timestep of 2 fs. The cutoffs for the short-range Lennard-Jones and electrostatic interactions were 1.0 nm. Long-range electrostatic interactions were computed using the smooth particle mesh Ewald method[61].

One H-REMD simulation of 30 μs (1.25 μs per scale, 24 scales) was performed for each covalent BLIP2 attachment site (azF41 and azF213) to the nanotube. H-REMD was necessary to achieve thorough sampling because standard MD simulations hardly sample different orientations of BLIP2 with respect to the nanotube (Supplementary Fig. 12). In H-REMD, we execute MD simulations in parallel for which the energy scales progressively reduce the attractive part of the Lennard-Jones interactions between BLIP2 and the carbon nanotube. Consequently, this improves the dissociation of BLIP2 from the nanotube at larger scaling, allowing it to sample an ensemble of orientations with respect to the nanotube as the system diffuses back to lower scaling, while preserving the overall structure of BLIP2. Specifically, there are 24 Hamiltonian scales ranging from 1 (no scaling, original energy) to 0 (no Lennard-Jones attraction between BLIP2 and the nanotube): 1.00, 0.95, 0.91, 0.87, 0.84, 0.81, 0.79, 0.76, 0.73, 0.71, 0.68, 0.66, 0.63, 0.60, 0.55, 0.50, 0.45, 0.40, 0.34, 0.28, 0.22, 0.15, 0.08 and 0. The scales were optimized to yield an average exchange rate of approximately 30% between scales 1 and 0.87, 55% between scales 0.87 and 0.60, and 30% between scales 0.60 to 0. The intermediate scales ensure a smooth transition from BLIP2 having many interactions with the nanotube (scales 1 to 0.87) to BLIP2 being mostly detached from the nanotube (scales 0.60 to 0). Exchanges between neighbouring scales are tried each 10 ps, alternating between even-odd and odd-even scales. All Hamiltonian scales start from a different orientation of BLIP2 with respect to the nanotube (Supplementary Fig. 13).

### Electrostatic potential simulations

The APBS software was used to solve the non-linear Poisson-Boltzmann equation[43] to determine the electrostatic potential generated by the protein on the SWCNT. The structures sampled in the Hamiltonian scales 1 to 7 for the H-REMD simulations were taken with a timestep of 100 ps in the 500–1250 ns interval (52 500 structures in total for each simulation) for electrostatic analysis. Statistics on the electrostatic potential were performed considering the weight of each structure as computed using the weighted histogram analysis method (WHAM)

(see SI for more details). The solvent was removed, keeping only BLIP2, the attachment and the SWCNT. To compute the effect of TEM-1 on the electrostatic potential generated on the nanotube, TEM-1 was added by aligning the structure of the BLIP2/TEM-1 complex onto the structure of BLIP2 in the frames sampled during the simulations. The aligned TEM-1 was kept along with the simulated BLIP2. For KPC-2, the same was done to obtain the BLIP2/KPC-2 complex as it shares high structural similarity with the BLIP2/TEM-1 complex according to AlphaFold predictions (Supplementary Fig. 2). The electrostatic potential calculations for BLIP2, BLIP2/TEM-1 and BLIP2/KPC-2 use the same parameters and box size (see SI for more details).

### Protein engineering and purification

The BLIP2 AzF variants were generated and purified as reported eleswhere[17,23], and plasmids containing the BLIP2-AzF variants are available from the authors on request. Briefly, site-directed mutagenesis was used to replace codons for the targeted amino acids with the amber stop codon TAG. The BLIP2-AzF mutants present in pET-BLIP2 vector (kindly provided by Tim Palzkill) together with AzF incorporation plasmid pEVOL-AzF (provided by Ryan Mehl via Addgene) were used to transform *E. coli* BL21 (DE3) for recombinant production; AzF was supplied exogenously in the culture medium. It is important to note that the production and purification of BLIP2 azF variants was performed under dark conditions where possible to minimise light exposure, and UV light sources were turned off during chromatography. BLIP2 AzF variants were purified after cell lysis by French Press using HisPur cobalt resin then concentrated and desalted using a PD10 column (Cytiva). Protein purity was assessed by SDS-PAGE.

TEM-1 was expressed using a pET-24a vector (a kind gift from the Makinen lab)[62] in *Escherichia coli* BL21(DE3) cultured in 2xTY media supplemented with kanamycin (30 mg/ml). On reaching an OD600 of -0.6, 1 mM IPTG was added to induce expression - kanamycin. The culture was then incubated at 22 °C for 16 hours. The next day, cells were pelleted at 5000 rpm and 5 °C for 20 minutes in the Fiberlite™ F9-6 × 1000 LEX rotor, and the supernatant removed. The pellet then underwent periplasmic extraction to carefully extract the protein, without lysing the whole. A similar procedure was used for KPC-2 except chloramphenicol (12.5 μg/ml) was used during cell culture. Periplasmic extraction is an outer membrane lysis technique used to purify β-lactamase proteins from the periplasmic space, without lysing the whole cell. The cell pellet from inoculated culture was resuspended in 30 mL periplasmic extraction buffer and stirred slowly at room temperature for 10 minutes. The cells were then re-pelleted by (50 mM Tris-HCl, pH 8, 20% (w/v) sucrose1 mM EDTA) centrifugation at 10000 x g at 4 °C for 10 minutes, in the Fiberlite™ F21-8 x 50 y rotor. Supernatant was discarded, and the cells resuspended in 30 mL chilled magnesium sulphate solution, being stirred slowly over ice for 10 minutes to release the periplasmic proteins. Next, the centrifugation step was repeated again, with the supernatant containing the soluble periplasmic proteins. The supernatant was then mixed in equal volumes with 50 mM Tris-HCl, pH 8 buffer in preparation for column chromatography.

A Resource Q anion exchange column was equilibrated with 30 mL dH₂O and 30 mL 50 mM Tris-HCl, pH 8, buffer. Next, the periplasmic supernatant (mixed 50:50 with water:Tris buffer) was loaded onto the column, with absorbance monitored at 280 nm. Protein was eluted from the column using a gradient of 0 to 1 M NaCl over 30 mL. Fractions containing the desired protein were identified and collected. Size exclusion chromatography was performed using a HiLoad™ 16/600Superdex™ S75 pg column. The column was equilibrated with 50 mM Tris-HCl, pH 8. The BL sample was concentrated in a 10 kDa molecular weight cut off column (Fisher Scientific) until the total protein volume was *circa* 1 mL. The protein sample was loaded with absorbance being monitored at 280 nm. All visible absorbance peaks were fractionated and analysed by SDS-PAGE.

## Device fabrication and protein attachment

The electrodes were patterned as conventional field-effect transistor structure with source and drain electrodes[17,63]. The patterned electrodes featured a nanosized gap of 300 nm between the source and drain electrodes. These patterned electrodes were fabricated and purchased from ConScience. Nanosized electrodes were fabricated on a p-doped Si/SiO₂ wafer by a combination of laser and electron beam lithography, followed by evaporating a thin adhesive layer of Cr and a thick layer of Au. 0.1 mg of 98% semiconducting single walled carbon nanotubes (SWCNTs, Sigma-Aldrich, product number 750522.

CAS number: 308068-56-6, 98% semiconducting, mixed chirality) was dispersed in 1 mL of 1% SDS solution via sonication for 1 h. The SWCNT sample was centrifuged for 1 h and the supernatant was collected and used as stock solution. To immobilise a small bundle of SWCNTs between electrodes, dielectrophoresis (DEP) was performed by applying an alternating current (AC) voltage between electrodes after SDS-dispersed SWCNT solution was cast on the electrodes. Typically, the frequency of the generator was switched onto Vp-p = 3 V at f = 400 kHz. After DEP, immobilised SWCNTs were characterised by AFM measurements. SDS was washed away with deionised water (DI), following the DEP immobilisation process: AFM imaging confirms this. Protein attachment was performed by applying UV light after a 1 μM BLIP2 protein solution was cast on the CNT-FETs. After 5 min of UV light exposure, the substrate was washed with water for 5 min to remove excess proteins from the substrate and blown gently with nitrogen gas. Topography analysis of the electrodes was done with a Bruker Dimension Icon atomic force microscope (AFM) with ScanAsyst Air tips. The images were analysed with NanoScope Analysis. Some devices were excluded if no SWCNTs were deposited, or the I-V measurements did not display ohmic behaviour. The choice of device functionalised with a BLIP AzF variant and the BL analysed with a particular device was random.

## Conductance measurements

Conductance measurements were obtained with a probe station (PS-100, Lakeshore) equipped with a semiconducting parameter analyser (Keithley, 4200 SCS) at room temperature. I-V curves were measured with bias sweeping mode (−1 V to 1 V) applied to source electrode while drain electrode was grounded. For real-time measurements, 100 mV source-drain bias was applied across the devices which have been covered with a drop of DPBS buffer solution. DPBS buffer was first added to the devices after reading of the current was stable, which cannot induce changes in current. This stable reading of the current through devices was recorded as background. Subsequently, 5 μL of TEM-1, KPC-2 or BSA solutions of various concentrations were cast on the devices. When present together with TEM-1, BSA was present at a final concentration of 0.5% (w/v). The resulting conductance profiles were collected by calculating $\Delta I/I_0$ ($\Delta I/I_0 = (I_{meas} - I_0) / I_0$, where $I_{meas}$ is the measured conductance after the addition of the target analyte, and $I_0$ is the measured conductance with only DPBS buffer solution added. The data was then fitted to a one-site total binding equation in Graph-Pad Prism 10.

### Reporting summary

Further information on research design is available in the Nature Portfolio Reporting Summary linked to this article.

## Data availability

Source data are provided with this paper. Modelling data and associated files can be found on Zenodo at https://doi.org/10.5281/zenodo.13149548. Protein structures used in this work include PDB ID 3qi0 [https://doi.org/10.2210/pdb3QI0/pdb]. Source data are provided with this paper.

## Code availability

Modelling was performed using open source software. Molecular dynamics was performed using Gromacs (https://www.gromacs.org). GROMACS 2019.6 was patched with PLUMED 2.6.2 (https://www.plumed.org) for the H-REMD simulations, and APBS 1.3 (https://apbs.readthedocs.io/en/latest/index.html) was used for the electrostatic simulations. Additional parameter files are available via GitHub[64].

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

## Acknowledgements

R.E.A.G., S.C. and C-S.L. contributed equally to this work. D.D.J would like to thank the EPSRC (EP/J015318/1 and EP/V048147/1) for supporting

this work. We gratefully acknowledge financial support from the U.S. Air Force Office of Scientific Research under award FA8655-21-1-7003. R.E.A.G. was supported by the Biotechnology and Biological Sciences Research Council-funded South West Biosciences Doctoral Training Partnership [training grant reference BB/M009122/1]. S.C. acknowledges support from the Program for College Research of the Fonds de recherche du Québec – Nature et technologie (FRQ-NT #275304 and #321076). K.R. was supported by Wellcome Trust Institutional Strategic Support Fund (grant reference AC1910IF14) awarded to D.D.J. S.C. and D.D.J would also like to acknowledge the Welsh Assembly Government-Quebec Government collaboration funding scheme for supporting this research. The authors would like to thank the Cardiff School of Biosciences Protein Technology Hub for helping with the production and analysis of proteins. The BLIP2, TEM-1 and KPC-2 containing plasmids was a kind gift from Prof Tim Palzkill. The pEVOL plasmid for incorporating azF was a kind gift from Ryan Mehl via Addgene (catalogue #164579). For the purpose of open access, the authors have applied a CC BY public copyright licence (UKRI permitted 'Open Government Licence') to any Author Accepted Manuscript version arising.

## Author contributions

R.E.A.G., S.C. and C.-S.L. contributed equally to the research. All authors contributed to the writing of the paper and analysing data. R.E.A.G. undertook initial rotamer modelling, produced protein and contributed to conductance measurements and analysis. S.C. undertook replica-exchange molecular dynamics simulations and electrostatic profile modelling. C.-S.L. prepared nanoscale electrodes and contributed to conductance measurements and analysis. H.M. undertook control conductance measurements. K.R. produced protein. M.P. conceived and directed the project, and contributed to data analysis. D.D.J. conceived and directed the project, and contributed to data analysis.

## Competing interests

The authors declare no competing interests.

## Additional information

**Peer review information** : *Nature Communications* thanks the anonymous, reviewer(s) for their contribution to the peer review of this work. A peer review file is available.

