## [Peer Review File · Nature Communications]

REVIEWER COMMENTS

Reviewer #1 (Remarks to the Author):

The authors present a study about detection of Beta-Lactamase using BLIP2 receptors covalently grafted to CNT-FETs. They provide extended modelling about the expected electrostatic gating effect as well as electrical measurements on several devices.

While I like the overall approach modelling of the effect combined with respective measurements and structural device characterization including the covalent attachment and find it certainly of high significance for field, I have to reject the manuscript in the current form for being published in Nature communications.

The selection of receptors and analytes appears well motivated as well as the argumentation about which orientations of the BLIP2 receptor matter. The idea is, to study the influence of the electrostatic environment on the transport channel of the CNTs.

The transport measurements, however, are not carried out in a way that allows for comparing them to the modelling.

First, the y-Axis labels of the current measurements are inconsistent and/or incorrect (Figure 2d and Figure 4a) - at least with regard to the definition given in the methods section (and there are no other definitions given).

Second, the authors find opposite behavior for BLIP2^{213azF} (current increase for TEM-1 analyte added) compared to a study published by some of the authors earlier (decrease of current after adding TEM-1 analyte in reference 17 in their manuscript). Though the BLIP2 grafting is done in a little bit a different way via pi-pi stacking with the Pyrene-DBCO-AzF is attached to the same position of the BLIP2. There is no indication that the orientation of the BLIP2 is different. This is not discussed.

I am not surprised that the results seem not to reproduce very well between the studies: Since the contacts to the devices are not passivated, it is impossible to distinguish between contact and channel effects in the measurements (compare, e.g., Chen et al, JACS 126, 1563-1568 (2004)). The interface between contacts and CNTs is difficult to control in general, so variations between devices are very common in different transport fields.

Furthermore, proteins tend to (partially) unfold due to hydrophobic interactions with CNTs (compare, e.g., Marchesan and Prato in Chem. Comm. 51, 4347 (2015)) and thus may change or lose their function. Therefore, the CNTs need to be wrapped or functionalized with materials (lipids, PEG, surfactants etc.) to prevent this from happening (e.g., Ravelli et al, RSC Advances 3, 13569-13582 (2013)). If the SDS used in the fabrication of the samples does create such an environment, it is not clear, how the functionalization with BLIP2 works in presence of the SDS molecules. And of course the SDS wrapping needs to be taken into account regarding the attachment orientation and regarding the electrostatic environment. All sketches though imply CNTs without wrapping as does the discussion of the AFM measurements. The functionalization of the CNTs happens with the same probability all around the circumference (as long as there are no steric clashes with the surface). Thus, some proteins can be quite close to the surface of the substrate. This might also have an impact on their integrity and surface treatment with PEG, lipids, etc. might be necessary.

Before transport experiments can be used to show the effect of protein binding on the transport channel seen in the modelling, the device fabrication needs to be thoroughly revised (passivation of contacts, check impact of hydrophobicity with according CNT/substrate surface treatment).

Reviewer #2 (Remarks to the Author):

In this sophisticated study, the authors demonstrate impressive control over the nano-architecture of carbon nanotube-based FETs. Gwyther et al. report how molecular modeling leads to the optimized presentation of an enzyme-binding protein on the CNT surface. This selective, covalent attachment, facilitated by the genetically introduced azido-phenylalanine, enhances the detection of two β -lactamases, as confirmed through experimental validation. The study is well-conducted, and the manuscript is well-written. The presentation and description of the data, including the supplementary information, enable readers to navigate through this interdisciplinary study. Overall, it represents a rational but significant advancement over their previous work. However, the novelty and implications for the FET/biosensor field need further elucidation. Therefore, I recommend publication following the revisions summarized below.

- 1) To emphasize the manuscript's novelty, the authors should contextualize their findings and improvements by comparing them to other FET designs. They should discuss whether the concept of side-specific protein attachment combined with systemic modeling has been previously employed to optimize other FET systems.
- 2) The authors should discuss the advantages of the covalently, photochemically attached binding protein over the previously described non-covalent pyrene anchor (doi.org/10.1002/anie.202104044).
- 3) The authors should elaborate on how the chemical design strategy can be easily applied to other systems. Is there a universal design principle summarizing the modeling approach? Can future designs of analyte-binding motifs be simplified, or is an elaborate workflow with comprehensive simulations necessary?

Technical Comments:

- a) How many binding units are present per CNT-FET device? What is the minimum number of binding units required for effective CNT-FET functionality? Additionally, can the introduced defect density be confirmed, and can the covalent SWCNT surface modification be validated through Raman spectroscopy?
- b) The authors should comment on the CNT-FET's performance in relation to the specific SWCNT material used and specify the type of SWCNTs employed for device fabrication. Could the SWCNT chirality influence the sensing performance or the covalent azido-phenylalanine photochemistry?
- c) The manuscript should clarify which SWCNT chirality was used for modeling and electrostatic potential simulations. The CNTs depicted in the presented Figures (e.g., S5, S6, S11) appear to have a zig-zag configuration, suggesting a metallic nature.
- d) Figure S4 and its implications would benefit from a more detailed description and discussion.

e) Figure S11 should be explained and referenced within the main manuscript.

f) The authors discuss the application of the developed biosensor for rapid sensing of antibiotic resistance in bacteria. To bolster this perspective, the performance of the BL-sensing device should be tested in the presence of potentially interfering molecules, such as those found in complex media or biofluids.

g) It would be advantageous for completeness if the authors evaluate the sensing performance of the BLIP2213azF-CNT-FET device after the addition of non-specific targets/proteins not expected to bind, such as denatured BL.

h) The authors should comment on the potential application of the developed biosensor. How can the testing for antibiotic resistance account for intracellularly produced BL.

i) The authors should elaborate on the physiological range of BL (e.g., in the relevant biofluid) and how the biosensor's performance aligns with it.

Reviewer #3 (Remarks to the Author):

To the editor and authors.

The paper by Gwyther, Côté and co-workers reports on a well-designed study that is an excellent integration of simulations and experiments. I will only comment on the simulations, which is my main expertise (I have read the experimental sections, and from my point of view, there are no concerns. The text is clear and, in my assessment, the data presented support the conclusions).

- What are the noteworthy results?

The detection of two BLs that are structurally related but give different conductance is noteworthy as BLs are of interest due to their role in anti-microbial resistance. The study is also a first step towards a more rational design of NT-based biosensors.

- Will the work be of significance to the field and related fields?

Yes. In my assessment, the paper reports a well-designed study that is an excellent integration of simulations and experiments and can serve as the basis for the future rational design of probes that are selective for a protein of interest. As noted in my comments below, the simulations are NOT predictive in their current form but are certainly well-designed, executed and analysed.

- How does it compare to the established literature? If the work is not original, please provide relevant references.

I cannot comment on this. I am not an expert in NTs and biosensors.

- Does the work support the conclusions and claims, or is additional evidence needed?

Yes, mostly. See my concerns about the simulations below. I have no concerns regarding the experimental point of view.

- Are there any flaws in the data analysis, interpretation and conclusions? - Do these prohibit publication or require revision?

See my comments below regarding the simulations.

- Is the methodology sound? Does the work meet the expected standards in your field?

See my comments below regarding the simulations.

- Is there enough detail provided in the methods for the work to be reproduced?

See my comments below regarding the simulations

My main concern relates to the REMD simulations, which are too short in my assessment based on the presented data. I think the paper is otherwise solid, demonstrates novelty, and is of significance to the field such that it can be published if this limitation is clearly acknowledged. Alternatively, the authors can extend the simulations.

There should also be more information on how the relative orientation was defined. See my comments further below.

Main concern related to sampling of orientations

The authors state, "The interval 500 to 750 ns of the unscaled MD (scale 1.0) was analyzed because the number of orientations remains stable on that interval with 2 main clusters for."

Stable implies this is the preferred orientation. However, how do the authors know these are not simply the orientations the protein is trapped in? The angle vs. time and angle populations clearly shows very limited sampling of orientations. The vast majority of orientations are never sampled (Fig S4).

In my view, there are now several ways to address this issue:

1. Do further analysis to show that this is truly a stable i.e. preferred orientation of the protein on the NT.

A first step would be to calculate the angle vs time and angle populations from all the scaled replicas.

Then also check what orientations 'trickle down' into the unscaled replica. If there is a wide range of orientations that trickle down and the protein then moves into the orientation that dominates the unscaled replica, then this would strengthen the statement that these are the stable conformation.

However, I doubt that 250ns is sufficient to sample enough orientations for this to be the case. The unscaled orientations will sample a much larger range, and even if that covers the entire accessible range ie produces an almost complete sampling of the angle population, then I doubt 250 ns is enough to get sufficient statistics of different orientations trickling down into the unscaled replica and rearranging. So then option 3 becomes relevant.

2. Extend the simulations. In my experience, even with scaled REMD simulations of at least 2-3 us are

required to get something resembling sufficient sampling or relative orientations. This is nothing intrinsic to the system under investigation here but simply a limitation when sampling the relative orientation of a protein/peptide on a surface such as an NT or membrane.

Alternatively, the authors could run three independent 1- μ s REMD simulations, starting from different structures (relative NT-protein orientations) in each replica. The system is not sampling enough if each simulation samples a different orientation. If certain orientations appear in all three simulations, then that would indicate that these orientations are truly stable /preferred conformations.

I appreciate that this is a lot of work and HPC resources. Still, in my view, the sufficient sampling of the relative orientations underpins the following statements from the main paper “Here we aimed to devise an accurate method to model the likely initial binding orientation of the receptor protein allowing us to estimate the distance and thus electrostatic influence between the incoming analyte protein and the SWCNT. BLIP2 will act as the receptor protein on the CNTs, for both its potential in antimicrobial resistance (AMR) diagnostics and to test our modelling approach. As BLIP2 binds a range of different BLs, this allowed us to model how different analyte proteins with distinctive electrostatic surfaces influence SWCNTs and thus, conductances.”

3. The authors can use the existing simulations with limited sampling and rewrite the relevant sections in the paper in a way that acknowledges this limitation. Such an edit would have to clearly point out that while current simulations are consistent with experimental data in terms of explaining how the different orientations/clusters led to different ESPs and thus can explain the different conductance, the simulations cannot be used to predict the most stable orientation or design modifications to alter selectivity or optimal attachment sites. For this, much longer simulations (likely $>5 \mu$ s are required). I also suggest removing the word ‘predict’ from the title and the manuscript in the context of the simulation.

Other comments

Introduction

“While TEM-1 and KPC-2 are structurally and functionally very similar, they have distinctive electrostatic surface profiles (Figure 1f). Given the importance of these proteins to this study, I think a bit more information is needed here. what is “very similar”? same overall fold, sequence ID or similarity? What are their functional differences? How different are the ESPs? At least on the image, I can’t see any large differences. Are there any specific areas on the protein surface that differ more than others? If the proteins are structurally similar, what makes their ESPs different?”

Results, main paper

“The average is performed on the electrostatic potential maps of 2500 sampled configurations in the converged interval.” Convergence of what property? And how was this convergence assessed?

Methods, supplementary material

Attachment parameters. “Finally, during the simulations, only the nitrogen atom was kept fixed to

maintain attachment with the SWCNT, while all the other atoms of the attachment were free to move.” If the attachment is covalently linked to the NT why does the N atom have to be kept fixed? And how was it kept fixed? With a position restraint? What force constant was used?

H-REMD. “For our system, we realized that BLIP2 hardly samples different orientations with respect to the nanotube when using standard MD simulations”. How was this assessed? I.e. how were the relative orientation of the attachment on the NT calculated? In Fig S4 angles vs time and distribution of angles are shown. How were these angles defined? I assume using two vectors. Which atoms were used to define the vectors that define these angles? How sensitive is this analysis to the choice of atoms selected?

Related to the two above comments. “As the remaining nitrogen atom of the phenyl azide is frozen during the simulation, we do not explicitly define bonded interactions between the nanotube and the attachment.” How would this affect the relative orientation the attachment can sample on the NT?

Fig S4. Caption. Please specify which simulation this analysis was obtained from and how many frames were used ie which part of the trajectory.

Electrostatic potential simulations. “This two-step procedure was necessary to accurately solve, in a reasonable time the electrostatic potential of the thousands of configurations generated by the HREMD simulations.” I suggest being more specific about how many structures were analysed and from which part of the simulation. The last 250 ns? I assume the non-scaled replica only.

Minor comments

Abstracts. SWCNTs, define abbreviation

HREMD section, supp material. “aren’t”. Contractions should not be used in scientific writing.

Responses to Reviewers' Comments

NCOMMS-23-36375A-Z

We wish to thank the Reviewers for their excellent critique. We have addressed all the comments and believe this has strengthened the manuscript. Our detailed responses to each comment can be found below.

It should be noted that additional figures added to the SI meant that we had to renumber the original SI figures in both the main manuscript and text. We have not highlighted these changes.

Reviewer #1

The authors present a study about detection of Beta-Lactamase using BLIP2 receptors covalently grafted to CNT-FETs. They provide extended modelling about the expected electrostatic gating effect as well as electrical measurements on several devices.

While I like the overall approach modelling of the effect combined with respective measurements and structural device characterization including the covalent attachment and find it certainly of high significance for field, I have to reject the manuscript in the current form for being published in Nature communications.

The selection of receptors and analytes appears well motivated as well as the argumentation about which orientations of the BLIP2 receptor matter. The idea is, to study the influence of the electrostatic environment on the transport channel of the CNTs.

The transport measurements, however, are not carried out in a way that allows for comparing them to the modelling.

We are glad the reviewer state they “*like the overall approach modelling of the effect combined with respective measurements and structural device characterization including the covalent attachment and find it certainly of high significance for field*”, and that “*the selection of receptors and analytes appears well motivated as well as the argumentation about which orientations of the BLIP2 receptor matter*”.

The reviewer adds “*The transport measurements, however, are not carried out in a way that allows for comparing them to the modelling*” and lists specific points needing clarification. We addressed below each and every issue raised by the Reviewer.

1) “*First, the y-Axis labels of the current measurements are inconsistent and/or incorrect (Figure 2d and Figure 4a) - at least with regard to the definition given in the methods section (and there are no other definitions given)*”.

We thank the reviewer for their comment. We changed y-Axis labels accordingly to the reviewer's suggestion. Figure 2d and 4a y-Axes is now $\Delta I/I_0$.

2) “*Second, the authors find opposite behavior for BLIP2^{213azF} (current increase for TEM-1 analyte added) compared to a study published by some of the authors earlier (decrease of current after adding TEM-1 analyte in reference 17 in their manuscript). Though the BLIP2 grafting is done in a little bit a different way via pi-pi stacking with the Pyrene-DBCO-AzF is attached to the same position of the BLIP2. There is no indication that the orientation of the BLIP2 is different. This is not discussed*”.

Reviewer 1 raises an important point, which highlights a missing aspect in the manuscript, namely that the attachment approach employed is important in defining the properties of bioFET and not just the

attachment position. By comparing our results here with our previous work, we can undertake such an evaluation. With regards to the reviewers' specific comment, we should make it clear that these are two very different strategies and will not result in equivalent effects despite attachment at the same residues - they are different in terms of the chemical immobilisation strategies employed (covalent vs non-covalent) and **crucially the nature (presence and absence) of extended flexible linkers**. In the study presented here, we employed a direct covalent attachment strategy: irradiation with near-UV light, generates a reactive nitrene radical on the phenyl azide group in the BLIP2 protein (through the precise designed incorporation of the non-canonical amino acid azF) resulting in **direct attachment to CNT sidewalls**. In the previously published manuscript (ref 17), we used pyrene to interface with the CNT sidewalls through non-covalent pi-pi stacking. Critically, the **pyrene is coupled to the protein via an 8-atom single bond linker** (that can sample different rotamer forms) to the **bulky dibenzylcyclooctene** which then reacts with the phenyl azide group in BLIP2 to form a trizole ring on strain-promoted azide-alkyne cycloaddition. In comparison, the direct phenyl azide photochemical attachment is essentially a **zero-length linker** with restricted rotamer sampling. The direct photochemical attachment will thus generate a closer, more intimate tethering of BLIP2 with a restricted set of orientations.

It is well known that linker design is very important in biomolecular conjugation and in biosensor design, with the linker systems having a major role in dictating sensor characteristics. Given the significant difference in linkers, it is reasonable to assume that the two strategies result in very different distances and allowed spatial arrangement of the BLIPs - and hence different orientation - with respect to the CNTs, resulting in distinct electrostatic environments presented on BL binding. Therefore, it is not surprising that the CNT-FET biosensing devices generated by the two different chemical attachment strategies exhibit divergent behaviour, i.e. current increase or decrease upon TEM analyte binding.

Following the reviewer's comment, we believe it is important to discuss and highlight how the actual attachment strategy involved is important for both modelling and implementation of biosensor designs so have now included a discussion aspect with respect to this (**p6, p14 and new SI Figure S1**).

3) *"I am not surprised that the results seem not to reproduce very well between the studies: Since the contacts to the devices are not passivated, it is impossible to distinguish between contact and channel effects in the measurements (compare, e.g., Chen et al, JACS 126, 1563-1568 (2004)). The interface between contacts and CNTs is difficult to control in general, so variations between devices are very common in different transport fields"*.

As per the point made in the answer to point 2 above, the two studies mentioned by the reviewer should not lead to the same results because of the very different chemical attachment strategies (covalent vs non-covalent) together with the linker length and its flexibility. The covalent attachment employed was chosen for the direct and more intimate tethering of the proteins to the CNTs, so avoiding a flexible linker granting greater control on the protein orientation.

There is no reproducibility issue when comparing the two studies: the reproducibility (or rather repeatability) of each separate study is solid, and the control experiments presented in both cases are convincing. In this regard, we do not experience significant variations between devices of the same type: see for example the standard deviations for the conductance measurements presented in Figures 2d and 4b-c. Moreover, the control experiments we performed (with both bare CNTs and with the BLIP protein immobilised in an orientation known to sterically block β -lactamases binding) show in both cases negligible change in conductance upon β -lactamases addition (see Figure S10).

We feel that our results are thus clear. It should be noted that because of the way we design and construct our devices, the electrophilic reactivity of the nitrene radical (generated on irradiation of the azF) means that BLIP2 will only permanently attach to the CNT channel. It should also be noted that our device

configuration exhibits fewer metal-nanotube contacts compared to the micromat design employed by Chen et al, which is likely to minimise the effect of non-specific irreversible adsorption on the measured conductance changes. Therefore, in our small channel gap devices (300 nm) a passivation of the contacts did not seem to be necessary. Indeed, we obtained consistent results among the same type of devices, and negligible response for both bare CNT devices and when the BLIP protein was immobilised in an unfavourable orientation for β -lactamases recognition (see again Figure S10).

4) *“Furthermore, proteins tend to (partially) unfold due to hydrophobic interactions with CNTs (compare, e.g., Marchesan and Prato in Chem. Comm. 51, 4347 (2015)) and thus may change or lose their function. Therefore, the CNTs need to be wrapped or functionalized with materials (lipids, PEG, surfactants etc.) to prevent this from happening (e.g., Ravelli et al, RSC Advances 3, 13569-13582 (2013)). If the SDS used in the fabrication of the samples does create such an environment, it is not clear, how the functionalization with BLIP2 works in presence of the SDS molecules. And of course the SDS wrapping needs to be taken into account regarding the attachment orientation and regarding the electrostatic environment. All sketches though imply CNTs without wrapping as does the discussion of the AFM measurements”.*

Protein function is reliant on a correctly folded protein. If a protein is unfolded it will not retain its function. In the case of BLIP2, its molecular structure determines its ability to recognise and bind the BLs. If BLIP2 was unfolded, it would lose its ability to bind to either KPC-2 or TEM-1 resulting in a signal comparable to that observed for our control mutant BLIP^{49azF} or the bare SWCNTs (Figure S10). **This is clearly not the case.** Plus, the adage “not all proteins” are equal is very much true with proteins exhibiting hugely varying thermodynamic, kinetic and thermal stability, with varying tolerance to hydrophobic environments. Thus, it cannot be assumed that every protein will unfold in the presence of CNTs or other nano-carbon materials (e.g. graphene). This is borne out experimentally in this paper, additional work from our groups (DOIs: 10.1002/adfm.202112374, 10.1002/anie.202104044, 10.1021/acs.bioconjchem.9b00719, 10.1039/c7ra11166e) and others (see our recent review DOI: 10.1002/cbic.202200282) and other reviews (e.g. DOI: 10.1021/ar300347d), which include many additional examples, including DOI: 10.1021/acs.jpcc.1c01589, 10.1126/science.1214824, 10.1039/c5cc03906a; 10.1021/ja311604j; 10.1021/nl304209p. Therefore, it should not be expected that every protein will unfold when attached to nano-carbon surfaces; the majority we work with remain functional and thus folded. Indeed, the review mentioned above states that it is non-specific non-covalent interactions that “may” [not always] cause a protein to unfold. We are not using non-specific non-covalent interactions here. **We have now added a sentence to the main manuscript to clarify that the proteins remain folded (p9). We have also added additional AFM data to provide evidence that BLIP2 retains BL binding capacity on attachment (Figure S4d-e and associated discussion in legend).**

Regarding the SDS wrapping of our CNTs, we wash the dispersant away with deionised water (DI), following the DEP immobilisation process: AFM imaging confirms this. **We have added this information in the Methods section of the manuscript (p 20).**

5) *“The functionalization of the CNTs happens with the same probability all around the circumference (as long as there are no steric clashes with the surface). Thus, some proteins can be quite close to the surface of the substrate. This might also have an impact on their integrity and surface treatment with PEG, lipids, etc. might be necessary”.*

While we cannot exclude that some proteins might be slightly closer to the SiO₂ surface, due for example to their attachment to the “side” of CNTs, our results suggest we have enough BLIP proteins that remain functional and are able to detect the incoming BLs; and this happens in a reproducible/repeatable way between the same kind of devices, suggesting a relatively consistent number of active BLIP2 proteins properly tethered to the CNTs (Figure 2d and 4a-c). Hence, even if

some proteins are closer to the surface and might be non-functional due to a compromised integrity, this must be minimal and not relevant enough to affect the biosensing response of our devices. Therefore, in our case a treatment with PEG or lipids was not necessary.

6. *“Before transport experiments can be used to show the effect of protein binding on the transport channel seen in the modelling, the device fabrication needs to be thoroughly revised (passivation of contacts, check impact of hydrophobicity with according CNT/substrate surface treatment).”*

Here the reviewer reiterates some of the points they made earlier, and that we addressed in responses number 3 and 4. Indeed, as mentioned in response 3, for our small channel gap devices (300 nm) a passivation of the contacts did not seem to be necessary. In support of this, we would like to stress that we obtained consistent results among the same type of devices, and negligible response for both bare CNT devices and when the BLIP protein was immobilised in an unfavourable orientation for β -lactamases recognition (see again Figure S10). With regards to the impact of hydrophobicity, as per our discussion in response 4 to this reviewer, protein function is reliant on a correctly folded protein. If a protein is unfolded due to hydrophobic interactions with the CNTs or the substrate surface, then the protein will not retain its function. In the case of BLIP2, its molecular structure determines its ability to recognise and bind the BLs. If BLIP2 was unfolded, it would lose its ability to bind to either KPC-2 or TEM-1 resulting in a signal comparable to that we observed for our control mutant BLIP^{49azF} or the bare SWCNTs (Figure S10). This is clearly not the case in our study. As per our discussion in response 3, it should not be expected that every protein will unfold when attached to nano-carbon surfaces; the majority we work with remain functional and thus folded. The reviews we mentioned (see for example our recent one DOI: 10.1002/cbic.202200282) state that it is non-specific non-covalent interactions that “may” -not always- cause a protein to unfold. We are not using non-specific non-covalent interactions here. We have now added a sentence to the main manuscript to clarify that the proteins remain folded (p9). We have also added additional AFM data to provide evidence that BLIP2 retains BL binding capacity on attachment (Figure S4d).

Reviewer #2

In this sophisticated study, the authors demonstrate impressive control over the nano-architecture of carbon nanotube-based FETs. Gwyther et al. report how molecular modeling leads to the optimized presentation of an enzyme-binding protein on the CNT surface. This selective, covalent attachment, facilitated by the genetically introduced azido-phenylalanine, enhances the detection of two β -lactamases, as confirmed through experimental validation. The study is well-conducted, and the manuscript is well-written. The presentation and description of the data, including the supplementary information, enable readers to navigate through this interdisciplinary study. Overall, it represents a rational but significant advancement over their previous work. However, the novelty and implications for the FET/biosensor field need further elucidation. Therefore, I recommend publication following the revisions summarized below.

We are please the Reviewer states that *“In this sophisticated study, the authors demonstrate impressive control over the nano-architecture of carbon nanotube-based FETs”* and that *“The study is well-conducted, and the manuscript is well-written”*. The Reviewer further adds *“Overall, it represents a rational but significant advancement over their previous work”*. The reviewer does recommend publication following revision addressing the points below.

1) *“To emphasize the manuscript's novelty, the authors should contextualize their findings and improvements by comparing them to other FET designs. They should discuss whether the concept of side-specific protein attachment combined with systemic modelling has been previously employed to optimize other FET systems”*.

To the best of our knowledge, we are not aware of other studies combining site-specific protein attachment and systemic modelling to optimize CNT-FET (carbon nanotube) or GFET (graphene) systems. We have now added some text in the introduction (**p. 4**) of the manuscript highlighting the novelty of our approach.

2) *“The authors should discuss the advantages of the covalently, photochemically attached binding protein over the previously described non-covalent pyrene anchor (doi.org/10.1002/anie.202104044).”*

We have now added text in the main manuscript briefly discussing the differences between the two distinct attachment approaches, and the advantage of the direct and less flexible protein attachment strategy employed in the work submitted here (**p6 and p14**). We have also added a new supporting Figure (S1) to clearly show the differences between the two attachment approaches.

3) *“The authors should elaborate on how the chemical design strategy can be easily applied to other systems. Is there a universal design principle summarizing the modelling approach? Can future designs of analyte-binding motifs be simplified, or is an elaborate workflow with comprehensive simulations necessary?”*

We have now expanded on our initial brief statement in the conclusions of our manuscript to address this (p16). Briefly, we expect our modelling approach to foster new studies quantitatively comparing different chemical design strategies (e.g. length, flexibility and site of the attachment) because our approach is applicable to CNT-FET or graphene-FET devices for any analyte-receptor complex. It will then be possible to identify universal chemical design strategies yielding greater device sensitivity.

The Reviewer further makes the following **Technical Comments** that we answer below:

a) *“How many binding units are present per CNT-FET device? What is the minimum number of binding*

units required for effective CNT-FET functionality? Additionally, can the introduced defect density be confirmed, and can the covalent SWCNT surface modification be validated through Raman spectroscopy”?

This is an excellent question and one that we have also considered. We were able to do such a calculation previously when analysing AFM images of single SWCNT systems (see DOI: 10.1002/adfm.202112374; 10.1021/acs.bioconjchem.9b00719). The use of SWCNTs bundles used here makes this task very difficult. To circumvent this, we have now analysed AFM images (including after biosensing on the addition of a BL) where we see single SWCNT protrusions (see **new addition to Figure S4**). We have measured via AFM imaging the average number of BLIP proteins per CNT length, and obtained the value of 3.6 ± 0.7 BLIP2s per 100 nm per individual visible CNT and added this to the text (see **p9**).

It is extremely challenging to control and verify the minimum number of binding units (BLIPs) required for effective CNT-FET functionality, and we think this is beyond the work presented here. Therefore, we do not expect such a major disruption to the SWCNT bond network as occurs for other processes such as oxidation. We have previously tried to determine the defect density on CNTs and graphene via Raman (see DOI: 10.1002/adfm.202112374; 10.1039/c7ra11166e), however the low number of defects (one per protein bound breaking a single pi bond) and the spatial resolution of the technique did not allow us to unequivocally confirm this.

b) “The authors should comment on the CNT-FET's performance in relation to the specific SWCNT material used and specify the type of SWCNTs employed for device fabrication. Could the SWCNT chirality influence the sensing performance or the covalent azido-phenylalanine photochemistry”?

In the methods section we should have better specified the carbon nanotubes employed in our study. We used mixed chirality single-walled carbon nanotube (Sigma-Aldrich, 98% semiconducting, mixed chirality). We have now added this information in the methods section of the manuscript. The different chiralities present can be identified from the UV-vis near-IR absorbance spectra we collected (see reviewer specific Figure 1 below). The covalent chemistry should not be significantly affected by the different chiralities present: the radical will react to any CNT as well as graphene or electron rich material. Studying the device performance as a function of the chirality is an interesting idea. Future studies could investigate the performance of CNT-FET sensing devices as a function of the chirality employed: current efforts in our laboratories indeed intend to clarify this for various biosensing moieties, but this represents a study by itself, beyond the scope of the manuscript presented here.

Reviewer specific Figure 1. UV-vis near-IR absorbance spectra of carbon nanotubes used in our study.

c) *“The manuscript should clarify which SWCNT chirality was used for modeling and electrostatic potential simulations. The CNTs depicted in the presented Figures (e.g., S5, S6, S11) appear to have a zig-zag configuration, suggesting a metallic nature”.*

The chirality of carbon nanotubes is not relevant in MD simulations because atoms are treated as point particles interacting through an empirical forcefield (e.g. AMBER). The chirality is also not important for electrostatic potential simulations because we are looking at the electrostatic potential generated by the protein on the surface of the nanotube. Here, we use a (10,0) nanotube –one of the major chirality forms present in the CNTs used (see spectra in answer point b above)- because the zig-zag configuration has a very short periodicity making it suitable for molecular dynamics (MD) simulation with periodic boundary conditions. This is now mentioned in the Methods section on page 17.

d) *“Figure S4 and its implications would benefit from a more detailed description and discussion”.*

We have now remade Figure S5 (which was Figure S4) following the reviewers’ comments. We now illustrate the definition of the two angles in Figure S5a along with their description in the caption. We also provide a supplementary discussion of this figure in its caption.

e) *“Figure S11 should be explained and referenced within the main manuscript”.*

We now refer to Figure S11 in the methods of the main manuscript and refer to the Supporting Information for more details.

f) *“The authors discuss the application of the developed biosensor for rapid sensing of antibiotic resistance in bacteria. To bolster this perspective, the performance of the BL-sensing device should be tested in the presence of potentially interfering molecules, such as those found in complex media or biofluids”.*

The reviewer makes a valid point, and it is something we have looked at previously and found the use of serum systems did not perturb BL sensing capability (see reference 17). Here we wanted to focus on how the modelling linked through to experiment so we did not undertake the equivalent of serum measurements; as the main difference with our previous studies with regards to device setup is the BLIP2 attachment process we would expect similar limited impacts of interfering molecules here. Indeed, the potentially major issue in device performance will be ionic strength and we performed the experiments here under physiologically relevant high salt conditions. We feel our existing control experiments will account for changes in conductance due to non-specific binding (Figures 4b and S10)

g) “It would be advantageous for completeness if the authors evaluate the sensing performance of the BLIP2213azF-CNT-FET device after the addition of non-specific targets/proteins not expected to bind, such as denatured BL”.

We feel that our experimental observations are clear for BLIP2^{213azF} – we see current changes dependent on the BL sampled and have suitable controls in place to detect non-specific binding (see Figures 4b and Figures S10). We have added a phrase at bottom p14/top of p15 to convey this point. We would not see such an observation if BL was unfolded as it would not be a conformation to bind BLIP2. Secondly, to keep the BL unfolded we will need to maintain conditions that promote the unfolded protein, including when added to biosensor (e.g. in the presence of chaotropic agent or heat). Such conditions are likely to induce unfolding of BLIP2 also so we will not be able to differentiate a null effect due to non-binding of the unfolded BL from receptor unfolding. Finally, unfolding the BL will lead to exposure of buried hydrophobics which can undergo non-specific adsorption on the SWCNT giving a false readout. Therefore, we think that our controls with the BLs added to the pristine SWCNTs devices and the use of a non-binding BLIP version (49azF) together with the distinct conductance characteristics of different BLs for the BLIP2^{213azF} devices are sufficient.

h) “The authors should comment on the potential application of the developed biosensor. How can the testing for antibiotic resistance account for intracellularly produced BL”.

For clarity, β -lactam antibiotics work by inhibiting bacterial cell wall synthesis so BLs need to be secreted to inactivate the antibiotic. BLs contain an N-terminal signal sequence that targets them for secretion out of the cell. Thus, BL site action is not intracellular. We have added a clarifying note regarding this on p4. We have also updated p16 in the Conclusion section to outline potential applications of the BL biosensor.

i) “The authors should elaborate on the physiological range of BL (e.g., in the relevant biofluid) and how the biosensor's performance aligns with it”.

After extensive analysis of the literature, to our knowledge the physiological range of BLs is not known as this will vary depending on bacterial type, BL type, site of infection, length of infection etc. As we state above, the paper is not about developing a sensor that for direct use in the clinic but to show how modelling can optimise underlying construction. That said our current devices are capable of sensing down to the fmole/ μ l level of BLs.

Reviewer #3

To the editor and authors.

The paper by Gwyther, Côté and co-workers reports on a well-designed study that is an excellent integration of simulations and experiments. I will only comment on the simulations, which is my main expertise (I have read the experimental sections, and from my point of view, there are no concerns. The text is clear and, in my assessment, the data presented support the conclusions).

- What are the noteworthy results?

The detection of two BLs that are structurally related but give different conductance is noteworthy as BLs are of interest due to their role in anti-microbial resistance. The study is also a first step towards a more rational design of NT-based biosensors.

- Will the work be of significance to the field and related fields?

Yes. In my assessment, the paper reports a well-designed study that is an excellent integration of simulations and experiments and can serve as the basis for the future rational design of probes that are selective for a protein of interest. As noted in my comments below, the simulations are NOT predictive in their current form but are certainly well-designed, executed and analysed.

- How does it compare to the established literature? If the work is not original, please provide relevant references.

I cannot comment on this. I am not an expert in NTs and biosensors.

- Does the work support the conclusions and claims, or is additional evidence needed?

Yes, mostly. See my concerns about the simulations below. I have no concerns regarding the experimental point of view.

- Are there any flaws in the data analysis, interpretation and conclusions? - Do these prohibit publication or require revision?

See my comments below regarding the simulations.

- Is the methodology sound? Does the work meet the expected standards in your field?

See my comments below regarding the simulations.

- Is there enough detail provided in the methods for the work to be reproduced?

See my comments below regarding the simulations

My main concern relates to the REMD simulations, which are too short in my assessment based on the presented data. I think the paper is otherwise solid, demonstrates novelty, and is of significance to the field such that it can be published if this limitation is clearly acknowledged. Alternatively, the authors can extend the simulations.

There should also be more information on how the relative orientation was defined. See my comments further below.

We are pleased that the reviewer finds that *“the paper by Gwyther, Côté and co-workers reports on a well-designed study that is an excellent integration of simulations and experiments”* and that it *“can serve as the basis for the future rational design of probes that are selective for a protein of interest”*.

The reviewer however raises concerns on the extent of the REMD simulations *“which are too short in [their] assessment based on the presented data”* to be predictive even if they *“are certainly well-designed, executed and analysed”* and even if the reviewer *“thinks the paper is otherwise solid, demonstrates novelty, and is of significance to the field”*. The reviewer then proposes ways to address

this.

We thank the reviewer for their insightful suggestions: we choose to extend our simulations and analyze them more thoroughly. We believe that this has strengthened the conclusions of our study, as demonstrated below in our answers to their comments.

Main concern related to sampling of orientations

The authors state, “The interval 500 to 750 ns of the unscaled MD (scale 1.0) was analyzed because the number of orientations remains stable on that interval with 2 main clusters for.” Stable implies this is the preferred orientation. However, how do the authors know these are not simply the orientations the protein is trapped in? The angle vs. time and angle populations clearly shows very limited sampling of orientations. The vast majority of orientations are never sampled (Fig S4).

We agree that the H-REMD simulations benefited a more thorough analysis (see following answers). Concerning Figure S5 (was S4), we now present the definition of the two orientation angles in panel a, and we show those that are accessible to BLIP2 (i.e. no clash with the nanotube) in panel c. This figure shows that the H-REMD simulations sample most of the accessible orientations when looking at all scales. For its part, the data in panel b shows the most stable orientations at scale 1 (unscaled energy) in terms of energy due to favorable protein-CNT interactions.

In the following, we address the reviewer’s suggestions to strengthen our study.

In my view, there are now several ways to address [the sampling] issue:

1. Do further analysis to show that this is truly a stable i.e. preferred orientation of the protein on the NT. A first step would be to calculate the angle vs time and angle populations from all the scaled replicas. Then also check what orientations ‘trickle down’ into the unscaled replica. If there is a wide range of orientations that trickle down and the protein then moves into the orientation that dominates the unscaled replica, then this would strengthen the statement that these are the stable conformation. However, I doubt that 250 ns is sufficient to sample enough orientations for this to be the case. The unscaled orientations will sample a much larger range, and even if that covers the entire accessible range ie produces an almost complete sampling of the angle population, then I doubt 250 ns is enough to get sufficient statistics of different orientations trickling down into the unscaled replica and rearranging. So then option 3 becomes relevant.

As suggested by the reviewer, we performed further analysis of the H-REMD simulations and we extended the simulations to 1250 ns per replica (extended by 500 ns) for a total of 30 μ s for each system.

First, the orientations sampled at all Hamiltonian scales along with the accessible orientations are now shown in Figure S5c. From that figure, we observe that the H-REMD simulations sample most of the accessible orientations, including those that are near the CNT. As expected, the orientations sampled at low scaling (scale 12 and smaller) are those for which the protein is near the CNT, while the orientations sampled at high scaling (scale 12 and greater) are those for which the protein is not in contact with the CNT because the attractive part of the Lennard-Jones interaction between the protein and the CNT are progressively scaled in the H-REMD simulations by a factor going from 1 (unscaled: scale number 1) to 0 (completely scaled: scale number 24). From Figure S5b, we observe the most energetically favorable orientations at the unscaled (real) energy, corresponding to a subclass of the orientations for which the protein is more tightly in contact with the CNT.

Second, following the observations from Figure S5c, we decided to combine the sampling of all scales to consider every possible orientation in our analysis. To do so, we used the Weighted Histogram Analysis Method (WHAM) to determine the weights at the unscaled (real) energy of every

configurations sampled at every scales, even those that are not energetically favorable enough to diffuse all the way to the unscaled replica. Using these weights, we determined the normalized population histogram of the orientations of BLIP2 with respect to the CNT to identify the orientations that are the most energetically favorable. We observe that both BLIP2^{41azF} and BLIP2^{213azF} mainly sample four clusters of orientation (Figure S6) resulting in different positions of TEM-1 with respect to the CNT (Figure S7).

Third, we analyzed the orientations sampled as a function of the replica number for all Hamiltonian scales (reviewer specific Figure 2, see below). At the unscaled energy, we observe the most energetically favoured orientations due to tighter interactions between the protein and the CNT (Panel 1). For BLIP2^{41azF}, we see that the two largest clusters – cluster 1 centered at 54°/130° with a population of 38% and cluster 2 centered at 50°/313° with a population of 31% (see Figure S6) – are sampled by different replica indexes, while the other less populous clusters – cluster 3 centered at 72°/137° with a population of 23% and cluster 4 centered at 62°/277° with a population of 8% – are mostly sampled by one replica index. Specifically, cluster 1 is mostly sampled by replica indexes 4 (74%), 20 (15%), 12 (8%) and 16 (1%); cluster 2 is mostly sampled by replica indexes 19 (46%), 8 (26%), 21 (18%), 13 (7%) and 3 (1%); cluster 3 is mostly sampled by replica index 1 (99%); and cluster 4 is sampled by replica index 9 (100%). For BLIP2^{213azF}, we also see that the two biggest clusters – cluster 1 centered at 44°/297° with a population of 49% and cluster 2 centered at 40°/248° with a population of 38% (see Figure S6) – as well as the smaller cluster 4 centered at 42°/119° with a population of 6% are sampled by different replica indexes, while the cluster 3 centered at 31°/30° with a population of 7% is mostly sampled by one replica index. Specifically, cluster 1 is mostly sampled by replica indexes 8 (60%), 9 (26%), 13 (6%) and 10 (3%); cluster 2 is mostly sampled by replica indexes 10 (67%), 13 (29%) and 9 (1%), 13 (7%); cluster 3 is mostly sampled by replica indexes 7 (95%), 4 (1%) and 16 (1%); and cluster 4 is sampled by replica indexes 6 (63%), 5 (6%), 14 (5%), 24 (5%), 15 (5%) amongst others. At the other Hamiltonian scales, we observe a more diverse set of replica indexes sampling the orientation clusters (Panels 4 to 7). Also, other less energetically favoured orientations for which the protein is not interacting with the CNT are sampled by a variety of replica indexes (left of Panels 10 to 24).

In summary, we see that the accessible orientations of BLIP2 with respect to the CNT, including those that are energetically more favorable because the protein interacts with the CNT, are sampled when looking at all scales (Figure S5c). Also, we see that the biggest clusters, at least, are visited by different replica indexes at low scaling (Panels 1 to 7 in Reviewer Specific Figure 2). Furthermore, each orientation cluster consists of a range of orientations, some more energetically favoured than others (Figures S5c vs S6).

Therefore, we believe that the main conclusion of our study can be maintained considering that we have extended the H-REMD simulations and that we used WHAM to include all configurations sampled at all scales for the analysis (Figures 3, S5, S6, S7 and S8). Namely, we can conclude from our results that the choice of the attachment site of BLIP2^{41azF} vs. BLIP2^{213azF} leads to a significantly different positioning of TEM-1/KPC-2 with respect to the CNT for all their orientation clusters (Figure S7), resulting in a significantly different electrostatic potential profile generated by the protein on the surface of the CNT (Figure 3), indicating a different response of the CNT-FET biosensor as confirmed by our experimental measurements (Figure 4).

Reviewer specific Figure 2. The replica numbers (colorbar) of all configurations sampled at every Hamiltonian scales (panels) for (a) BLIP2^{41azF} and (b) BLIP2^{213azF}.

2. Extend the simulations. In my experience, even with scaled REMD simulations of at least 2-3 us are required to get something resembling sufficient sampling or relative orientations. This is nothing intrinsic to the system under investigation here but simply a limitation when sampling the relative orientation of a protein/peptide on a surface such as an NT or membrane.

Alternatively, the authors could run three independent 1- μ s REMD simulations, starting from different structures (relative NT-protein orientations) in each replica. The system is not sampling enough if each simulation samples a different orientation. If certain orientations appear in all three simulations, then that would indicate that these orientations are truly stable /preferred conformations.

I appreciate that this is a lot of work and HPC resources. Still, in my view, the sufficient sampling of the relative orientations underpins the following statements from the main paper “Here we aimed to devise an accurate method to model the likely initial binding orientation of the receptor protein allowing us to estimate the distance and thus electrostatic influence between the incoming analyte protein and the SWCNT. BLIP2 will act as the receptor protein on the CNTs, for both its potential in antimicrobial resistance (AMR) diagnostics and to test our modelling approach. As BLIP2 binds a range of different BLs, this allowed us to model how different analyte proteins with distinctive electrostatic surfaces influence SWCNTs and thus, conductances.”

As suggested by the reviewer, we have extended our simulations to 1250 ns per replica (500 ns more) for a total time of 30 μ s for each system. As discussed in the previous point, Figure S5c shows that the H-REMD simulations sample most of the accessible orientations, including those for which the protein can interact with the CNT. Furthermore, our analysis now considers all configurations sampled at all Hamiltonian scales to identify the most energetically stable orientation clusters (Figure S6). While further extending the simulations could yield adjustments in the relative population of the orientation clusters for each system, we believe that the differences between BLIP2^{41azF} to BLIP2^{213azF} will be maintained since they are significant enough in terms of the orientation of BLIP2 with respect to the CNT (Figure S6), the position of TEM-1/KPC-2 with respect to the CNT (Figure S7) and the electrostatic potential generated by the protein on the CNT (Figure 3).

We would like to point out key differences between our study compared to other studies of protein/peptide on a surface that might explain faster sampling.

First, the protein-surface system investigated in our study is not as flexible as other systems. On the protein side, the structure of BLIP2 is known from X-ray crystallography and it experiences only minor structural changes when binding TEM-1 and KPC-2 β -lactamases. On the surface side, the surface of a CNT is homogeneous and very stable on the nanometer scale. Moreover, in our case, the biomolecule is covalently attached to the surface using a short linker (4-azido-L-phenylalanine). Therefore, the system studied here has less conformational freedom compared to other systems having a very flexible biomolecule (e.g. a ssDNA wrapping around a CNT) or a very heterogeneous and dynamic surface (e.g. a peptide interacting with a phospholipid membrane).

Second, in our case, we used H-REMD simulations in which we selectively scale the attractive interactions between the biomolecule and the surface, allowing to specifically target the relevant degrees of freedom impeding the sampling of the orientations of BLIP2 with respect to the CNT. Our approach is different from other H-REMD techniques such as REST2 for which both the attractive and repulsive terms of the Lennard-Jones biomolecule-biomolecule (full scale) and biomolecule-solvent (square root scale) interactions are scaled. While appropriate to simulate biomolecular folding, REST2 was not sufficient to detach BLIP2 from the nanotube even at very high scaling (preliminary tests not shown). Our approach is also different from other REMD techniques such as T-REMD for which the temperature of the system is different from one scale to the other. Again, high enough temperatures could not be reached to detach BLIP2 from the nanotube without unfolding BLIP2. Only the type of the REMD method used in our study allowed us to sample a spectrum of orientations that progressively converge toward the most energetically favoured orientations (Figures S5 and S6).

3. The authors can use the existing simulations with limited sampling and rewrite the relevant sections in the paper in a way that acknowledges this limitation. Such an edit would have to clearly point out that while current simulations are consistent with experimental data in terms of explaining how the

different orientations/clusters led to different ESPs and thus can explain the different conductance, the simulations cannot be used to predict the most stable orientation or design modifications to alter selectivity or optimal attachment sites. For this, much longer simulations (likely >5 us are required). I also suggest removing the word 'predict' from the title and the manuscript in the context of the simulation.

We have acknowledge the reviewer's suggestion and removed the word "predict" from the title and in relevant sections of the manuscript. We instead focus on the usefulness of these simulations to guide the conception of CNT-FET biosensors and to support the interpretation of the conductance signal measured experimentally (see the new Conclusion).

Other comments

Introduction

"While TEM-1 and KPC-2 are structurally and functionally very similar, they have distinctive electrostatic surface profiles (Figure 1f). Given the importance of these proteins to this study, I think a bit more information is needed here. what is "very similar"? same overall fold, sequence ID or similarity? What are their functional differences? How different are the ESPs? At least on the image, I can't see any large differences. Are there any specific areas on the protein surface that differ more than others? If the proteins are structurally similar, what makes their ESPs different?"

The protein belongs to the same structural class of BL, namely the class A serine BLs (mentioned on p4). It is hard to graphically show the full electrostatic surface in a static picture and we feel we have picked a common orientation for KPC-2 and TEM-1 to illustrate the differences in the electrostatic profile in 1f. We feel that the differences in red and blue (corresponding to negatively and positively charged patches, respectively) is clear in 1f – the figure highlights an area on each BL where the electrostatic surface differs. The 3D structures of TEM-1 and KPC-2 are similar with a C α RMSD of 0.81 Å (now mentioned on **page 4**). They have different number of charged residues. Specifically, TEM-1 has a total charge of -7 (36 negatively and 27 positively charged residues) and KPC-2 has a total charge of -1 (26 positively and 25 positively charged residues) – the number of charged residues has now been added to the manuscript (**p 12**). Consequently, their solvent accessible surface has a different electrostatic potential profile, which we think is clearly shown in Figure 1f.

Results, main paper

"The average is performed on the electrostatic potential maps of 2500 sampled configurations in the converged interval." Convergence of what property? And how was this convergence assessed?"

We added the section "Analysis of the H-REMD simulations" in the Supporting Information (p. 4). We refer to that section of the Supporting Information in the Methods. In brief, we monitored the structural properties of BLIP2 (secondary and tertiary structures) and its interaction with the carbon nanotube (distance, orientation and number of contacts) over time. Using these metrics, the analysis interval was chosen to be from 500 to 1250 ns. The configurations considered for electrostatic potential analysis were taken in that interval with a 100-ps timestep.

Methods, supplementary material

Attachment parameters. "Finally, during the simulations, only the nitrogen atom was kept fixed to maintain attachment with the SWCNT, while all the other atoms of the attachment were free to move." If the attachment is covalently linked to the NT why does the N atom have to be kept fixed? And how was it kept fixed? With a position restraint? What force constant was used?"

We froze the N atom and the C atom to which the nanotube is linked (added on p. 1 in SI). This was necessary to reproduce the geometry of the known [2+1] cycloaddition reaction, whereby the nitrene radical formed on exposure to UV light inserts perpendicular to the SWCNT (Ref 6 in SI).

H-REMD. “For our system, we realized that BLIP2 hardly samples different orientations with respect to the nanotube when using standard MD simulations”. How was this assessed? I.e. how were the relative orientation of the attachment on the NT calculated? In Fig S4 angles vs time and distribution of angles are shown. How were these angles defined? I assume using two vectors. Which atoms were used to define the vectors that define these angles? How sensitive is this analysis to the choice of atoms selected?

We now better present the definition of these angles visually and in text (Figure S5a). Briefly, the vector used to calculate these angles goes from the center of the nanotube where the attachment site is to the center-of-mass of BLIP2. The center-of-mass is calculated from the positions of the carbon alpha atoms because the tridimensional structure of BLIP2 is stable throughout the simulations, as indicated by the low backbone RMSD over time (0.14 ± 0.02 nm for BLIP2^{41azF} and 0.10 ± 0.01 nm for BLIP2^{213azF}). The vector is projected on two planes to get the two angles: (1) the angle around the nanotube is defined on the plane perpendicular to the axis of the nanotube, and (2) the angle around the attachment site is defined on the plane perpendicular to attachment. Note that we have slightly changed our definition of the angle around the nanotube: we use the center of the nanotube instead of the attachment site on the nanotube as it is easier to interpret. Therefore, we changed the values in the text and in the figures accordingly, without affecting our previous observations.

Concerning the standard MD simulations, we performed a 1000 ns simulation for each system prior to using H-REMD (Figure S12). We realized that BLIP2 hardly samples different orientations with respect to the CNT in these simulations. This prompted us to use H-REMD to get a thorough and unbiased characterization of the orientations of BLIP2.

Related to the two above comments. “As the remaining nitrogen atom of the phenyl azide is frozen during the simulation, we do not explicitly define bonded interactions between the nanotube and the attachment.” How would this affect the relative orientation the attachment can sample on the NT?

Semi-empirical *ab initio* computation suggest that this type of functionalization is perpendicular to the carbon nanotube (Ref 6 in SI). Freezing nitrogen atom linked to the nanotube and the last carbon atom of the phenyl ring is sufficient to maintain this perpendicular orientation. Since the other atoms of the attachment are unrestrained, its aromatic ring and its joint with the backbone of BLIP2 can freely sample any local conformations allowed by the bonded interactions of the attachment (i.e. similar to a phenylalanine).

Fig S4. Caption. Please specify which simulation this analysis was obtained from and how many frames were used ie which part of the trajectory.

The caption of Figure S5 (which was Figure S4) is now showing the required information.

Electrostatic potential simulations. “This two-step procedure was necessary to accurately solve, in a reasonable time the electrostatic potential of the thousands of configurations generated by the HREMD simulations.” I suggest being more specific about how many structures were analysed and from which part of the simulation. The last 250 ns? I assume the non-scaled replica only.

We added the section “Analysis of the H-REMD simulations” in the Supporting Information (p. 4). We refer to that section of the Supporting Information in the Methods. In brief, we used the configurations sampled on the 500 to 1250 ns interval with a 100-ps resolution (7500 configurations per scale). We focused on the configurations sampled in the first seven Hamiltonian scales because they have the

greatest normalized weights ($>10^{-9}$ compared to a maximum of 10^{-5}).

Minor comments

Abstracts. SWCNTs, define abbreviation HREMD section, supp material. "aren't". Contractions should not be used in scientific writing.

We have corrected these.

REVIEWER COMMENTS

Reviewer #1 (Remarks to the Author):

The authors addressed all my points. Still, I have one minor issue.

I have still doubts about the detection mechanism. Especially for short channelled, Schottky-barriers can play an important role. The barrier height can change with changes in the electrostatic environment. So without passivation of the contacts, one cannot be sure that the detection is happening via changes in the channel.

Of course, this is also a valid detection mechanism.

The statistics of the devices do not counteract this. On the contrary: since the standard deviation increases with Blip attached, it seems clear that something is happening. Either at the channel or at the contacts.

I thus recommend that the conclusion about the detection mechanism should be a bit more cautious.

I suggest that the manuscript is published after addressing this point.

Reviewer #2 (Remarks to the Author):

The authors have made significant improvements, addressing most of the remaining points. However, the sensor concept still lacks some important controls, as outlined previously. I recommend publication after resolving these issues.

f) "The authors discuss the application of the developed biosensor for rapid sensing of antibiotic resistance in bacteria. To bolster this perspective, the performance of the BL-sensing device should be tested in the presence of potentially interfering molecules, such as those found in complex media or biofluids".

The reviewer makes a valid point, and it is something we have looked at previously and found the use of serum systems did not perturb BL sensing capability (see reference 17). Here we wanted to focus on how the modelling linked through to experiment so we did not undertake the equivalent of serum measurements; as the main difference with our previous studies with regards to device setup is the BLIP2 attachment process we would expect similar limited impacts of interfering molecules here. Indeed, the potentially major issue in device performance will be ionic strength and we performed the experiments here under physiologically relevant high salt conditions. We feel our existing control

experiments will account for changes in conductance due to non-specific binding (Figures 4b and S10)

The authors present a novel, modeling-driven sensor design, which needs to be thoroughly characterized to function as a biosensor. While previous biosensor concepts were not affected, the performance of the BLIP2213azF-CNT-FET device should be tested similarly. Please refer to point g for further details.

g) "It would be advantageous for completeness if the authors evaluate the sensing performance of the BLIP2213azF-CNT-FET device after the addition of non-specific targets/proteins not expected to bind, such as denatured BL".

We feel that our experimental observations are clear for BLIP2213azF – we see current changes dependent on the BL sampled and have suitable controls in place to detect non-specific binding (see Figures 4b and Figures S10). We have added a phrase at bottom p14/top of p15 to convey this point. We would not see such an observation if BL was unfolded as it would not be a conformation to bind BLIP2. Secondly, to keep the BL unfolded we will need to maintain conditions that promote the unfolded protein, including when added to biosensor (e.g. in the presence of chaotropic agent or heat). Such conditions are likely to induce unfolding of BLIP2 also so we will not be able to differentiate a null effect due to non-binding of the unfolded BL from receptor unfolding. Finally, unfolding the BL will lead to exposure of buried hydrophobics which can undergo non-specific adsorption on the SWCNT giving a false readout. Therefore, we think that our controls with the BLs added to the pristine SWCNTs devices and the use of a non-binding BLIP version (49azF) together with the distinct conductance characteristics of different BLs for the BLIP2213azF devices are sufficient.

I respectfully disagree with the authors' evaluation. Similar to the previous point, Figure 4b and S10 demonstrate the enhanced sensor response of the BLIP2213azF-conjugate, whereas a non-binding version or the bare SWCNTs do not show significant changes. Whether the BLIP2213azF-configuration is selective or not is not clear. The hypothesis "We would not see such an observation if BL was unfolded as it would not be a conformation to bind BLIP2" should be experimentally verified.

"Secondly, to keep the BL unfolded we will need to maintain conditions that promote the unfolded protein, including when added to biosensor"

Do the authors have literature supporting this explanation?

Presenting suitable control experiments of the BLIP2213azF-FET detecting its target BL in the presence of potentially interfering compounds (point f) and non-responding to non-specific targets (point g) will undoubtedly strengthen the manuscript.

i) "The authors should elaborate on the physiological range of BL (e.g., in the relevant biofluid) and how the biosensor's performance aligns with it".

After extensive analysis of the literature, to our knowledge the physiological range of BLs is not known as this will vary depending on bacterial type, BL type, site of infection, length of infection etc. As we state above, the paper is not about developing a sensor that for direct use in the clinic but to show how modelling can optimise underlying construction. That said our current devices are capable of sensing down to the fmole/ μ l level of BLs.

Given that the authors have already gathered this information, they should at least discuss one scenario (bacteria type, BL, infection) where the detection of fmole/ μ l of BLs could be relevant or of interest.

Reviewer #3 (Remarks to the Author):

The authors have addressed my concerns and I recommend the paper to be published in its revised form

NCOMMS-23-36375C response to reviewers.

We thank the reviewers for their largely positive comments and their recommendation to accept our manuscript on some additional revisions. The main aspect was the addition of control experiments, which we have now added to the paper (see additions to the main manuscript, SI and Figure S10). We have also updated the list of authors to include Haosen Miao who performed the additional experimental work. We have addressed each point below (in red below) and now hope that the paper is considered acceptable for publication in Nature Communications.

REVIEWER COMMENTS

Reviewer #1 (Remarks to the Author):

The authors addressed all my points. Still, I have one minor issue.

I have still doubts about the detection mechanism. Especially for short channelled, Schottky-barriers can play an important role. The barrier height can change with changes in the electrostatic environment. So without passivation of the contacts, one cannot be sure that the detection is happening via changes in the channel.

Of course, this is also a valid detection mechanism.

The statistics of the devices do not counteract this. On the contrary: since the standard deviation increases with Blip attached, it seems clear that something is happening. Either at the channel or at the contacts.

I thus recommend that the conclusion about the detection mechanism should be a bit more cautious.

I suggest that the manuscript is published after addressing this point.

Response to reviewer 1.

As reviewer pointed out, electrostatic gating and Schottky-barriers modification can both play a role in the conductance changes of CNT-FET sensing devices. Therefore, we investigated the conductance changes of BLIP2-modified CNT-FETs upon the addition of BSA (Bovine Serum Albumin), which can potentially modify Schottky-barriers via non-specific adsorption.

As illustrated in Figure S10c, our new experimental additions show no significant changes in conductance upon the addition of BSA, suggesting that potential Schottky-barrier modifications may not be a significant factor influencing the conductance changes in our system. To further assess this, an additional new experiment assessed BL sensing performance of our devices containing excess BSA (Figure S10d). The conductance of BLIP2-modified CNT-FET biosensors increased upon addition of BL, with a reasonable standard deviation, indicating that BSA is unlikely to significantly affect the sensing response of our devices.

Therefore, it is reasonable to assume that an electrostatic gating effect, rather than any Schottky-barrier effect, is the major sensing mechanism in our detection platform. Future research efforts, beyond the scope of this manuscript, will focus on a more detailed understanding of the different detection mechanisms that can contribute to the signal generation of CNT-FET based biosensors. We have now added the aforementioned control experiments to the supporting information of our manuscript (Figure S10c-d). Moreover, we have added some additional text at the end of the results section (p15) to cautiously stress what we have reason to believe is the main sensing mechanism responsible for the signal changes we see in our CNT-FET devices, namely electrostatic gating induced by the specific recognition of BLs by BLIP tethered to SWCNTs.

Reviewer #2 (Remarks to the Author):

As the reviewer intertwined their response with previous comments, we have tried to break down each aspect, with the recent comments highlighted in yellow.

The authors have made significant improvements, addressing most of the remaining points. However, the sensor concept still lacks some important controls, as outlined previously. I recommend publication after resolving these issues.

Reviewers original comment: f) “The authors discuss the application of the developed biosensor for rapid sensing of antibiotic resistance in bacteria. To bolster this perspective, the performance of the BL-sensing device should be tested in the presence of potentially interfering molecules, such as those found in complex media or biofluids”.

Our original response. *The reviewer makes a valid point, and it is something we have looked at previously and found the use of serum systems did not perturb BL sensing capability (see reference 17). Here we wanted to focus on how the modelling linked through to experiment so we did not undertake the equivalent of serum measurements; as the main difference with our previous studies with regards to device setup is the BLIP2 attachment process we would expect similar limited impacts of interfering molecules here. Indeed, the potentially major issue in device performance will be ionic strength and we performed the experiments here under physiologically relevant high salt conditions. We feel our existing control experiments will account for changes in conductance due to non-specific binding (Figures 4b and S10)*

The authors present a novel, modeling-driven sensor design, which needs to be thoroughly characterized to function as a biosensor. While previous biosensor concepts were not affected, the performance of the BLIP2213azF-CNT-FET device should be tested similarly. Please refer to point g for further details.

Reviewer original point. g) “It would be advantageous for completeness if the authors evaluate the sensing performance of the BLIP2213azF-CNT-FET device after the addition of non-specific targets/proteins not expected to bind, such as denatured BL”.

Our original response. *We feel that our experimental observations are clear for BLIP2213azF – we see current changes dependent on the BL sampled and have suitable controls in place to detect non-specific binding (see Figures 4b and Figures S10). We have added a phrase at bottom p14/top of p15 to convey this point. We would not see such an observation if BL was unfolded as it would not be a conformation to bind BLIP2. Secondly, to keep the BL unfolded we will need to maintain conditions that promote the unfolded protein, including when added to biosensor (e.g. in the presence of chaotropic agent or heat). Such conditions are likely to induce unfolding of BLIP2 also so we will not be able to differentiate a null effect due to non-binding of the unfolded BL from receptor unfolding. Finally, unfolding the BL will lead to exposure of buried hydrophobics which can undergo non-specific adsorption on the SWCNT giving a false readout. Therefore, we think that our controls with the BLs added to the pristine SWCNTs devices and the use of a non-binding BLIP version (49azF) together with the distinct conductance characteristics of different BLs for the BLIP2213azF devices are sufficient.*

I respectfully disagree with the authors' evaluation. Similar to the previous point, Figure 4b and S10 demonstrate the enhanced sensor response of the BLIP2213azF-conjugate, whereas a non-binding version or the bare SWCNTs do not show significant changes. Whether the BLIP2213azF-configuration is selective or not is not clear. The hypothesis "We would not see such an observation if BL was unfolded as it would not be a conformation to bind BLIP2" should be experimentally verified.

Our original response: “Secondly, to keep the BL unfolded we will need to maintain conditions that promote the unfolded protein, including when added to biosensor” “Such conditions are likely to induce unfolding of BLIP2 also so we will not be able to differentiate a null effect due to non-binding of the unfolded BL from receptor unfolding”

Do the authors have literature supporting this explanation?

Presenting suitable control experiments of the BLIP2213azF-FET detecting its target BL in the presence of potentially interfering compounds (point f) and non-responding to non-specific targets (point g) will undoubtedly strengthen the manuscript.

Response to reviewer.

We have now added two additional conductance experiments using the BLIP2^{213azF} CNT-FET devices with BSA (Bovine Serum Albumin) as the non-specific protein target. BSA is a commonly used standard protein to detect for non-specific effects, including in biosensors. The first experiment concerns conductance in the presence of varying concentration of BSA (new Figure S10c) – no change on conductance is observed over all the BSA concentrations used. The second experiment was performed to detect TEM-1 with an excess of BSA: the concentration conductance profile matches that of the TEM-1 alone (Figure 1S0d). We have also updated the text in the main manuscript (p15) to convey these new results and modified the experimental section to include these new experimental components (p20-21). **We now feel these experiments satisfy the reviewer's comments.**

The use of denatured BL is not possible due to basic concept in biochemistry that is the basis for 1972 Nobel prize in chemistry for Christian Anfinsen (<https://www.nobelprize.org/prizes/chemistry/1972/anfinsen/facts/>) and his pivotal experimental evidence of spontaneous protein refolding on removal from denaturant - indeed without the basic concept that a protein spontaneously folds, life as we know it would not exist. There are also papers that show BLs specifically quickly refold when removed from denaturing conditions (e.g. DOIs: 10.1021/bi701927y; 10.1002/prot.340220204; 10.1021/bi972143c; 10.1016/j.ijbiomac.2010.09.009)! This is why this experiment is not feasible.

Secondly, we do not need to experimentally verify that unfolded BL cannot bind BLIP2 as evidence already exist - the molecular structure of the complex is one example as BLIP2 makes direct contact with different sequence regions of BL: these contact regions are only meaningful in the context of the 3D structure of the BL as BLIP2 blocks the active site through multiple contact sites with BL (see diagram below). This is re-enforced by many biochemical studies, including with different BLs and their mutants (e.g. DOI: 10.1074/jbc.M804089200; 10.1074/jbc.M111.265058; 10.1074/jbc.M113.463521; 10.1002/pro.2505). The idea of molecular complementarity based on 3D structure is a common one and well utilised by biology. Indeed, as with the spontaneous folding concept, biology as we know would not exist without molecular complementarity based on 3D structural arrangement. Here, we are not looking at simple antibody- epitope-like binding event where the epitope can be a short linear sequence of amino acids. **We have now added a sentence to the introduction to make this clear (p4).**

Protein-protein interactions that define the BLIP2-TEM-1 complex

Author original comment. i) "The authors should elaborate on the physiological range of BL (e.g., in the relevant biofluid) and how the biosensor's performance aligns with it".

Original response. After extensive analysis of the literature, to our knowledge the physiological range of BLs is not known as this will vary depending on bacterial type, BL type, site of infection, length of infection etc. As we state above, the paper is not about developing a sensor that for direct use in the clinic but to show how modelling can optimise underlying construction. That said our current devices are capable of sensing down to the fmole/ μ l level of BLs.

Given that the authors have already gathered this information, they should at least discuss one scenario (bacteria type, BL, infection) where the detection of fmole/ μ l of BLs could be relevant or of interest.

Response to reviewer

As we said originally, there is no known physiological range of BL in infection - we looked at the literature and there was no answer. This is likely to be due to way bacterial infections and resistance are currently diagnosed - mostly through qualitative cell culturing (i.e. does the bacteria grow on a particular antibiotic or not and determining values such as minimum inhibitory concentration of a particular antibiotic) or through genomic approaches rather – neither of these provide direct measurement of BL levels. May be this is one question we can address in the future building on the work and the results shown in this paper.

Reviewer #3 (Remarks to the Author):

The authors have addressed my concerns and I recommend the paper to be published in its revised form

Response to reviewer. Thank you.

REVIEWERS' COMMENTS

Reviewer #2 (Remarks to the Author):

I recommend publishing the paper in its current form.

NCOMMS-23-36375C response to reviewers.

REVIEWER COMMENTS

Reviewer #1 (Remarks to the Author):

We were informed that the reviewer has withdrawn from the process but a request was made by the editor:

“Additionally, and as discussed over email, due to the lack of a Reviewer with the correct expertise to assess the newly added discussions regarding the detection mechanism (on page 15 of the manuscript file), we must ask that these are removed, and that any claims regarding mechanism are toned down, as previously suggested by Reviewer #1.

We have now done this (on Page 10 of the Word document) including adding the following to tone down the claims regarding the mechanism:

“While we cannot entirely exclude that Schottky-barrier modification 2, potentially induced by non-specific protein adsorption, may play a role in our system, it is reasonable to assume that an electrostatic gating effect induced by the specific recognition of BLs by BLIP tethered to SWCNTs, is significantly contributing to the sensing response of our detection platform.”

We have also changed “electrostatic gating” in the manuscript (highlighted red word below).

P4 of Word Document: “...where the electrostatic **influence** is greatest”

We hope this is now sufficient.

Reviewer #2 (Remarks to the Author):

“I recommend publishing the paper in its current form.”

No response needed.